# Leveraging Conditional Dependence for Efficient World Model Denoising

**Shaowei Zhang**[1,2], **Jiahan Cao**[1,2], **Dian Cheng**[1,2], **Xunlan Zhou**[3], **Shenghua Wan**[1,2], **Le Gan**[1,2], **and De-Chuan Zhan**[†1,2]

[1]National Key Laboratory for Novel Software Technology, Nanjing University, China
[2]School of Artificial Intelligence, Nanjing University, China
[3]School of Intelligence Science and Technology, Nanjing University, China
{zhangsw, caojh, chengd, wansh}@lamda.nju.edu.cn, wyattzhouxl@smail.nju.edu.cn,
{ganl, zhandc}@nju.edu.cn

## Abstract

Effective denoising is critical for managing complex visual inputs contaminated with noisy distractors in model-based reinforcement learning (RL). Current methods often oversimplify the decomposition of observations by neglecting the conditional dependence between task-relevant and task-irrelevant components given an observation. To address this limitation, we introduce *CsDreamer*, a model-based RL approach built upon the world model of ***Collider-structure Recurrent State-Space Model (CsRSSM)***. CsRSSM incorporates colliders to comprehensively model the denoising inference process and explicitly capture the conditional dependence. Furthermore, it employs a decoupling regularization to balance the influence of this conditional dependence. By accurately inferring a task-relevant state space, CsDreamer improves learning efficiency during rollouts. Experimental results demonstrate the effectiveness of CsRSSM in extracting task-relevant information, leading to CsDreamer outperforming existing approaches in environments characterized by complex noise interference. [1]

## 1 Introduction

Reinforcement Learning has achieved remarkable success in complex applications such as autonomous driving [1] and conversational interactions [2, 3]. A key factor in these advancements is the agent's ability to accurately perceive and interpret observations from its environment. However, real-world observations are often corrupted by noise and extraneous information, which can severely impair an agent's ability to make sound decisions. This challenge intensifies significantly when dealing with high-dimensional observations like images [4, 5, 6]. Therefore, effectively managing such noisy observations is essential for developing robust and reliable RL agents.

From a generative model perspective [7], at timestep $t$, an observation $o_t$ is generated by two distinct sets of latent variables: task-relevant variables $s_t$, which directly influence the agent's rewards or actions, and task-irrelevant variables $c_t$, which introduce noise and distractions without contributing to the task (Figure 1 (**a**)). Existing model-based RL approaches have sought to extract task-relevant information from observations [8, 9]. Some methods utilize two separate dynamics models to extract task-relevant and task-irrelevant information from observations [10, 11], while others decompose observations into task-relevant and task-irrelevant branches [12] or pursue more fine-grained decomposition [13, 14]. However, these methods oversimplify the decomposition of observations by neglecting the conditional dependence between $s_t$ and $c_t$ given $o_t$. The importance of this conditional dependence can be illustrated with a simple example: consider random variables

---

†: Corresponding author.
[1]The code is available at `https://github.com/Zhang-Shaowei/CsDreamer`.

39th Conference on Neural Information Processing Systems (NeurIPS 2025).

$A$, $B$, and $C$ where $A + B = C$. If $C$ is unknown, $A$ and $B$ are independent. However, once $C$ is observed (e.g., $C = c$), their joint probability becomes $P(a, b \mid C = c) = \frac{P(a,b)\mathbb{I}(a+b=c)}{P(C=c)}$.

This implies that if we determine $A = a$, then $B$ must be $c - a$. Thus, $A$ and $B$ become conditionally dependent given $C$. This conditional dependence can be leveraged for efficient denoising. For instance, as illustrated in Figure 1 (**b**), previous approaches perform separate inferences for $s_t$ (the walker's state) and $c_t$ (the background) by assuming conditional independence. However, the background information can significantly influence the inference of the walker's state. By capturing the conditional dependence between $s_t$ and $c_t$, we can sig-

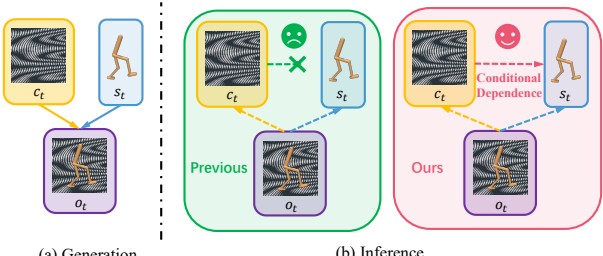

(a) Generation   (b) Inference

Figure 1: (**a**) Generation process of noisy observations. (**b**) Inference process where we consider the conditional dependence.

nificantly enhance the inference process, as the walker's state can be rapidly deduced by comparing differences between the background and the noisy observation. Therefore, capturing this conditional dependence is beneficial and can improve the inference process.

Inspired by this, we propose *Collider-structure Recurrent State-Space Model* (CsRSSM), a world model designed for denoising from a generative modeling perspective. CsRSSM extends Recurrent State-Space Model (RSSM) [15] by modeling the noisy observation generation process as a sequential collider-structure model. By explicitly utilizing the collider structures, CsRSSM exploits the conditional dependence between $s_t$ and $c_t$ given $o_t$ to effectively extract task-relevant information. Specifically, CsRSSM first infers the task-irrelevant variable $c_t$ from $o_t$ and then utilizes both $c_t$ and $o_t$ to infer the task-relevant latent variable $s_t$. This sequential inference approach utilizes the contextual information provided by task-irrelevant variables to enhance the efficiency of task-relevant information extraction. Additionally, we regularize the model using conditional mutual information to balance the conditional dependence between $s_t$ and $c_t$ during learning. Furthermore, existing methods typically employ masking mechanisms [10, 12] or tailored optimization objectives [16], which depend on prior knowledge or assumptions, to distinguish task-relevant information from task-irrelevant information. Compared with these strategies which may constrain model flexibility and introduce biases, CsRSSM adopts a generative modeling perspective, employing dual latent variables to model the generation of noisy observations and the environment dynamics. It relies solely on the decoupling regularization and the inherent optimization objectives of the generative model, without incorporating any additional prior knowledge or supplementary optimization objectives. Experimental results demonstrate the feasibility and effectiveness of this approach.

Finally, we introduce *CsDreamer*, an extension of the Dreamer framework [17], that employs CsRSSM to effectively extract task-relevant information from noisy observations. By conducting rollouts exclusively within the space of $s_t$, CsDreamer enhances the agent's ability to efficiently learn policies in high-dimensional visual environments characterized by complex noise distractors. Our contributions are threefold:

- We propose CsRSSM, a novel generative model for environment dynamics that explicitly utilizes conditional dependence to model the noisy observation generation.
- We introduce CsDreamer, which builds upon CsRSSM to effectively denoise observations and train policies within the task-relevant space.
- We conduct extensive experiments demonstrating that our approach outperforms existing state-of-the-art methods in handling noisy observations.

## 2 Preliminaries

**Recurrent State-Space Model.** A Recurrent State-Space Model (RSSM) [15] can capture the dynamic relationships in the latent space. Dreamer [18, 19, 17] adopts a world model based on RSSM to construct the dynamics model of the environment perceived by the agent in the latent space. In the real environment, the agent receives an observation input $o_t$ and performs an action $a_t$, after which it receives the next observation $o_{t+1}$ following the environment's state transition governed by the probability distribution $P(o_{t+1}|o_t, a_t)$. The world model captures the dynamics in observation space based

on a dynamics model in the latent space $\mathcal{Z}$, which primarily comprises the following components: the trajectory history model $h_t = f_\phi(h_{t-1}, z_{t-1}, a_{t-1})$, the encoder model $q_\phi(z_t|h_t, o_t)$, the transition model in latent space $p_\phi(\hat{z}_t|h_t)$ and the reward model $p_\phi(\hat{r}_t|h_t, z_t)$, where $\phi$ represents the parameter vector of the world model. The dynamics model in $\mathcal{Z}$ uses the same action space $\mathcal{A}$ as the real environment. In the dynamics model, $h_t$ records the historical trajectory information, $q_\phi(z_t|h_t, o_t)$ extracts the useful information from the observation, and $p_\phi(\hat{z}_t|h_t)$ represents the transition in the latent space.

## 3  Related Work

**Reinforcement Learning with Noisy Observations.**  Many recent studies have focused on the reinforcement learning problem with noisy observations. Some studies adopt metric-based representation learning approaches to deal with noisy observations [20, 21, 22, 23]. MIIR [24] adopts information-theoretic principles to learn invariant representations. InfoPower [8] prioritizes action-correlated factors. RePo [6] encourages reward-predictive yet compact encodings. Another line of work relies on data augmentation to reduce distractors [25, 26, 27]. Additionally, structured latent variables have been explored for denoising. SEAR [28] partitions agent- and environment-related components via segmentation, while DEAR [29] applies a similar mask-based approach without reconstructing the entire observation. Other methods decompose observations by training two separate dynamics models to separate task-relevant and task-irrelevant information [10, 11], splitting them into distinct branches [12], or adopting more fine-grained decomposition [13, 14]. However, these decomposition-based approaches typically assume conditional independence between factors given the observation.

**Generative Models in Reinforcement Learning.**  Generative models have played a pivotal role in tackling high-dimensional observations for RL. Some works employ the Variational Autoencoder (VAE) [30, 31] to learn lower-dimensional latent variables from images, facilitating latent-space policy optimization [32, 15]. Flow-based generative models [33, 34] also provide a way to learn flexible distributions with explicit likelihoods, though balancing model complexity with real-time efficiency can be challenging in RL settings. Diffusion models [35, 36] have demonstrated impressive performance in image generation, and their potential for RL has begun to attract attention [37]. However, most of these generative approaches focus primarily on high-fidelity reconstruction or predictive modeling, as opposed to explicitly separating out different observational factors such as background noise or irrelevant distractors.

**Disentanglement in Reinforcement Learning.**  Several studies concentrate on disentanglement within reinforcement learning to derive efficient representations for behavioral learning. TED [38] and CMID [39] introduce disentanglement techniques into feature learning process for RL. Our method draws inspiration from them while focusing on decoupling between two sets of latent variables rather than disentangling the original feature into individual feature dimensions. Some causal reinforcement learning approaches target the disentanglement of different state components [40, 41, 16], while they pay less attention to the observation generation process or employ counterfactual and intervention mechanisms [42, 43]. Instead, we treat noisy observations from a generative model perspective and focus on conditional dependence between task-relevant and task-irrelevant variables during inference. Although Cao et al. [44] also notice the conditional dependence, they combine two separate generative models rather than utilizing a unified model directly.

## 4  Method

In this section, we first introduce the underlying assumption regarding the structure of noisy observations and then present the *Collider-structure Recurrent State-Space Model* (CsRSSM) from a generative modeling perspective in Section 4.1. Subsequently, we introduce a decoupling regularization according to the characteristics of the network in Section 4.2 to facilitate subsequent reinforcement learning. We present the overall loss objective for the CsRSSM world model in Section 4.3 and introduce the policy learning in Section 4.4. The overall framework of the CsRSSM world model is shown in Figure 2.

### 4.1  Collider-Structure Recurrent State-Space Model

In real-world scenarios, agent observations are often contaminated by noise and irrelevant information, which degrades decision-making performance. From a generative modeling perspective, we propose the following assumption to elucidate the relationship between latent variables and the generation of observations in such environments.

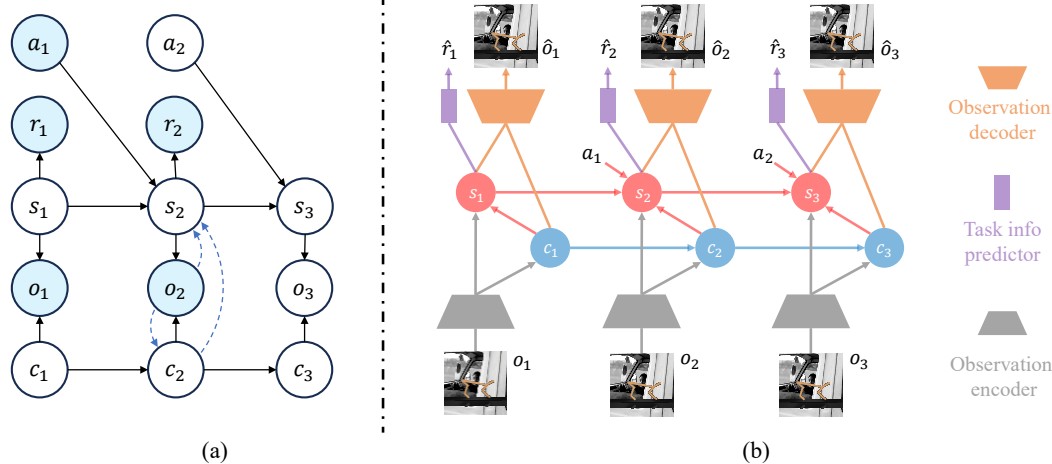

(a)                                               (b)

Figure 2: Framework of the CsRSSM world model. For brevity, content related to historical information in Eq. (3) is omitted. (**a**) Probabilistic graphical model of CsRSSM. Shaded nodes represent observed variables such as actions and observations. Among the unshaded nodes, $a_2$ and $o_3$ are future variables that have not yet been observed at timestep $t = 2$, while the remaining unshaded nodes are latent variables. The black solid lines represent the generative model, and the blue dotted lines illustrate the inference process, which accounts for conditional dependence. (**b**) The training framework of the world model.

**Assumption 4.1.** *(**Collider structure assumption**) There exist two distinct sets of latent variables that jointly determine the generation of the observation $o_t \in \mathcal{O}$: the task-relevant latent variable $s_t \in \mathcal{S}$, which is associated with the agent's actions and rewards, and the task-irrelevant latent variable $c_t \in \mathcal{C}$, which pertains solely to interference.*

This assumption posits that the observations an agent perceives are a composite of two underlying factors and acknowledges the conditional dependence between the latent variables $s_t$ and $c_t$ given the observation $o_t$. This nuanced modeling ensures that the interplay between task-relevant and task-irrelevant factors is accurately captured, thereby enhancing the agent's ability to make informed and robust decisions despite the presence of noise and irrelevant information. Under Assumption 4.1, we can obtain the following generative process

$$p(s_t, c_t, o_t) = p(c_t)p(s_t)p(o_t|c_t, s_t), \tag{1}$$

which is similar to the former methods [10, 12, 13]. Due to the presence of collider structures, conditional independence typically does not hold during inference; that is, $p(c_t, s_t|o_t) \neq p(c_t|o_t)p(s_t|o_t)$. Instead, we recognize the dependence between $s_t$ and $c_t$ and employ the chain rule of probability rather than the arbitrary factorization based on conditional independence to factorize the inference probability. There are two ways to implement the probability chain rule: either by inferring $c_t$ first or inferring $s_t$ first. In scenarios where agents receive noisy observations, the provision of additional task-irrelevant information can aid in inferring task-relevant information given the observations of an agent, as indicated by a reduction in conditional entropy: $\mathcal{H}(s_t|c_t, o_t) \leq \mathcal{H}(s_t|o_t)$. Therefore, we utilize

$$p(c_t, s_t|o_t) = p(c_t|o_t)p(s_t|c_t, o_t), \tag{2}$$

where the main difference lies in the inclusion of $c_t$ for the inference of $s_t$. We first use the observation $o_t$ to infer $c_t$, the task-irrelevant information, and then utilize both $c_t$ and $o_t$ to infer $s_t$, the task-relevant information.

To better capture the temporal information in noise, we model the task-irrelevant variables $c_t$ as time-varying. Constant noise can be regarded as a special case of time-varying noise where $c_t$ remains constant. In practice, the noise at the next moment is primarily influenced by the noise at the current moment. Hence, we posit the following assumption.

**Assumption 4.2.** *The transitions of the task-irrelevant latent variables $c_t$ satisfy the Markov property, i.e., $c_t \sim p(c_t|c_{<t}) = p(c_t|c_{t-1})$.*

**Remark 4.3.** *Assumption 4.2 is analogous to the Markov property in Reinforcement learning, with the distinction that transitions of $c_t$ are independent of the agent's actions.*

Consider the sequences $\{o_t, a_t, r_t, x_t\}_{t=1}^T$, where $t$ is the timestep, $a_t$ is the action taken by the agent, $r_t$ is the reward signal, $x_t$ is the episode continuation flag and $T$ is the length of the sequence. Under Assumption 4.1 and Assumption 4.2, we propose the CsRSSM world model, which leverages the conditional dependence of the collider structures to model the noisy observation generation process from a generative modeling perspective:

$$
\begin{cases}
\text{Task-irrelevant history:} & h_t^c = f_\phi(h_{t-1}^c, c_{t-1}) \\
\text{Task-relevant history:} & h_t^s = f_\phi(h_{t-1}^s, s_{t-1}, a_{t-1}) \\
\text{Task-relevant transition:} & p_\phi(\hat{s}_t|h_t^s) \\
\text{Task-irrelevant transition:} & p_\phi(\hat{c}_t|h_t^c) \\
\text{Task-irrelevant encoder:} & q_\phi(c_t|h_t^c, h_t^s, o_t) \\
\text{Task-relevant encoder:} & q_\phi(s_t|h_t^c, h_t^s, c_t, o_t) \\
\text{Reconstruction model:} & p_\phi(\hat{o}_t|h_t^s, s_t, h_t^c, c_t) \\
\text{Reward model:} & p_\phi(\hat{r}_t|h_t^s, s_t), \\
\text{Continuation predictor:} & p_\phi(\hat{x}_t|h_t^s, s_t)
\end{cases}
\tag{3}
$$

where $\phi$ is the parameter vector of the world model. Analogous to the RSSM [15], our approach incorporates historical information. We denote the task-irrelevant history as $h_t^c$ and the task-relevant history as $h_t^s$. The transition models provide the transition priors: $p_\phi(\hat{s}_t|h_t^s)$ predicts the next task-relevant state based on the current task-relevant history, while $p_\phi(\hat{c}_t|h_t^c)$ predicts the next task-irrelevant state from the current task-irrelevant history. The two observation encoders operate similarly to the transition models but incorporate observations as inputs.

The inclusion of historical information introduces conditional dependence, and the posterior distributions of $s_t$ and $c_t$ capture all conditional dependence between task-relevant and task-irrelevant information within the collider structures given the observations. Specifically, for each collider structure, the model first uses the history and the current observation $o_t$ to infer the task-irrelevant information $c_t$. Subsequently, it utilizes the history along with both $o_t$ and $c_t$ to infer the task-relevant information $s_t$. We use the features of both $c_t$ and $s_t$ to reconstruct the observation. In contrast, we only use the features of $s_t$ to predict the task information, such as rewards and episode continuation flags. These task information reconstructions facilitate the learning of task-relevant latent variables $s_t$. Finally, we derive the evidence lower bound (ELBO) loss for the CsRSSM world model from a generative modeling perspective:

$$
\begin{aligned}
\mathcal{L}_{\text{ELBO}} = -\sum_{t=1}^T \Big[ & \mathbb{E}[\ln p_\phi(o_t|h_t^s, s_t, h_t^c, c_t) + \ln p_\phi(r_t|h_t^s, s_t) + \ln p_\phi(x_t|h_t^s, s_t)] \\
& - \mathbb{E}[\text{KL}[q_\phi(c_t|h_t^c, h_t^s, o_t)\|p_\phi(c_t|h_t^c)]] - \mathbb{E}[\text{KL}[q_\phi(s_t|h_t^c, h_t^s, c_t, o_t)\|p_\phi(s_t|h_t^s)]] \Big],
\end{aligned}
\tag{4}
$$

where we maximize the log-likelihood of observations, rewards and continuation flags. The two KL divergence losses serve separately for task-relevant and task-irrelevant information, simultaneously training the priors toward the representations and regularizing the representations toward the priors. The derivation of Eq. (4) can refer to Appendix A.1. Notably, when faced with noisy observations, the presence of conditional dependence mitigates trivial solutions where $c_t = \mathbf{0}$ and $s_t = o_t$, ensuring that both $c_t$ and $s_t$ are effectively utilized.

## 4.2 Decoupling Regularization in CsRSSM

Directly training the CsRSSM world model may present some issues. Specifically, since the model training primarily relies on the reconstruction objectives, there is a risk of conflating the task-relevant and task-irrelevant variables during the training process. This conflation can hinder the model's ability to accurately disentangle the underlying factors of the observations, ultimately degrading performance and slowing convergence. To mitigate these issues, we propose explicitly decoupling the task-relevant latent variables $s_t$ from the task-irrelevant latent variables $c_t$. We

measure the information shared between $s_t$ and $c_t$ given $o_t$ using the conditional mutual information, denoted as $I(s_t; c_t|o_t)$. Noting the relationship that the KL divergence between conditional dependence and conditional independence is equivalent to the conditional mutual information, i.e., $\text{KL}[p(c_t|o_t)p(s_t|c_t, o_t)\|p(c_t|o_t)p(s_t|o_t)] = I(s_t; c_t|o_t)$, we achieve decoupling by introducing a regularization loss that minimizes the conditional mutual information between $s_t$ and $c_t$ given the condition variables $\{o_{\leq t}, a_{<t}\}$, denoted as $I(s_t; c_t|h_t^c, h_t^s, o_t)$. Minimizing this mutual information encourages the task-relevant and task-irrelevant variables to capture distinct and non-overlapping aspects of the observations, thereby facilitating more effective learning for CsRSSM. Directly minimizing the conditional mutual information is intractable. By incorporating $q_\xi(s_t|h_t^c, h_t^s, o_t)$, a variational distribution with parameters $\xi$ that approximates the true posterior, we can obtain an upper bound loss for the conditional mutual information:

$$\mathcal{L}_{\text{MI}} = \mathbb{E}[\text{KL}[q_\phi(s_t|c_t, h_t^c, h_t^s, o_t)\|q_\xi(s_t|h_t^c, h_t^s, o_t)]], \tag{5}$$

where $q_\phi(s_t|c_t, h_t^c, h_t^s, o_t)$ is the task-relevant encoder of CsRSSM. The Derivation can refer to Appendix A.2. We can minimize the conditional mutual information for the task-relevant latent variables $s_t$ and the task-irrelevant latent variables $c_t$ by minimizing the upper bound in Eq. (5).

## 4.3 Overall Objective for CsRSSM

Integrating the discussions in Sections 4.1 and 4.2, we obtain the final loss function for CsRSSM world model

$$\mathcal{L}_{\text{CsRSSM}} = \mathcal{L}_{\text{ELBO}} + \lambda\mathcal{L}_{\text{MI}}, \tag{6}$$

where the second item serves as the regularization of the ELBO loss to balance the conditional dependence between $s_t$ and $c_t$. The hyperparameter $\lambda$ controls the weight of the regularization. A larger $\lambda$ enforces greater separation between $s_t$ and $c_t$, promoting distinct representations of task-relevant and task-irrelevant information within the latent space, while a smaller $\lambda$ leverages the conditional dependence to facilitate more efficient inference.

Previous studies operate under the conditional independence assumption for $s_t$ and $c_t$ [10, 12] given an observation. This assumption is a special case of our formulation where the regularization coefficient $\lambda$ approaches infinity. In such scenarios, $\mathcal{L}_{\text{MI}}$ dominates the loss function, effectively reducing the mutual information between $s_t$ and $c_t$ to zero. This enforces strict independence between the task-relevant and task-irrelevant variables given an observation, aligning with the conditional independence assumption. Our approach generalizes existing methods by introducing a finite $\lambda$, enabling a flexible trade-off for the degree of decoupling enforced between $s_t$ and $c_t$. This extension incorporates the conditional dependence between $s_t$ and $c_t$ for world model denoising. More experimental discussions are provided in Section 5.3.

Moreover, existing methods typically employ masking mechanisms [10, 12] or tailored optimization objectives [16], which depend on prior knowledge or assumptions, to distinguish task-relevant information from task-irrelevant information. However, these strategies may constrain model flexibility and introduce biases. Instead, we adopt a generative modeling perspective, employing dual latent variables to model the generation of noisy observations and the environment dynamics. The model is trained solely based on the inherent optimization objectives of the generative model (the reconstruction errors and the prior regularizations in Eq. (4)), and the decoupling regularization according to the model network (Eq. (5)). It does not incorporate any additional prior knowledge or supplementary optimization objectives.

## 4.4 Policy Learning in CsDreamer

Finally, we propose *CsDreamer*, which is an extension of Dreamer [17] built upon CsRSSM. The agent's policy is learned during the imagination phase of the CsRSSM. By focusing on task-relevant information, we only utilize $s_t$ to choose the next action. i.e., $a_t \sim \pi_\theta(a_t \mid h_t^s, s_t)$, where $\pi$ is the policy of the agent. During the imagination phase, the agent interacts solely with the task-relevant transition model, $p_\phi(\hat{s}_t|h_t^s)$, to generate rollout trajectories, with rewards predicted by the reward model $p_\phi(\hat{r}_t|h_t^s, s_t)$. We predict the value utilizing $V_\psi(h_t^s, s_t)$ based on $s_t$. The agent policy is trained analogously to Dreamer [17]. Focusing on task-relevant information and leveraging the strengths of model-based reinforcement learning ensure efficient and targeted policy development.

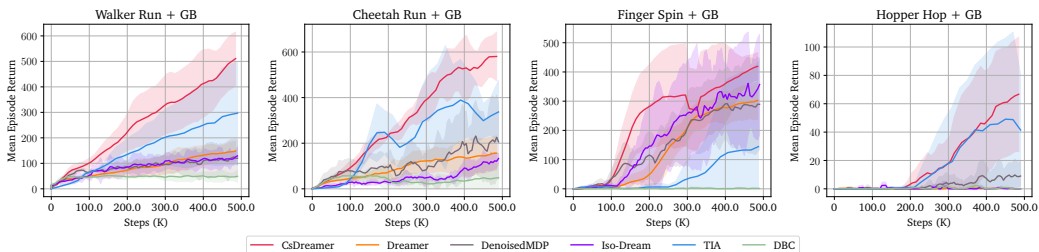

Figure 3: Performance on DMC using gray natural videos as background.

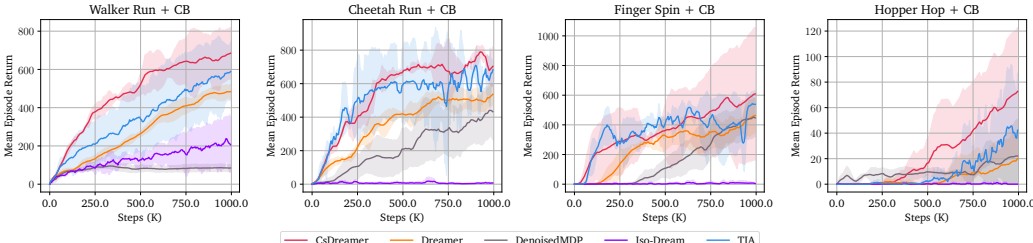

Figure 4: Performance on DMC using colorful natural videos as background.

## 5 Experiments

We begin by describing the experimental setup in Section 5.1. Subsequently, we address the following research questions: (1) Can CsDreamer enhance the training performance in complex environments with distractors? (Section 5.2); (2) What role does the decoupling regularization loss play in the overall objective, and how do the proposed modules affect performance? (Section 5.3); (3) Do the latent variables in the proposed framework exhibit interpretable semantics upon reconstruction visualization? (Section 5.4).

### 5.1 Experiment Setup

**Benchmarks.** We evaluate our model and baselines on visual control tasks. First, we assess performance on four DeepMind Control Suite (DMC) [45] tasks: *Walker Run*, *Cheetah Run*, *Finger Spin* and *Hopper Hop*. To introduce noise distractors, we replace the original backgrounds with two types of task-irrelevant information. The first is a **gray** background composed of natural videos from the Kinetics 400 dataset [46], following the DBC [21] configuration (denoted as **GB**). The second is a **colorful** background derived from DAVIS 2017 videos [47], adhering to the background distractor settings in Distracting Control Suite [48] (denoted as **CB**). These benchmarks require the agent to extract task-relevant information, identify the target entity within the DMC environment, and effectively filter out background distractions. Subsequently, we evaluate these approaches in the more realistic simulated driving environment, CARLA [49]. Here, the agent must extract task-relevant information from visual perception while mitigating distractions such as trees and dynamic sunlight. We also conduct experiments on the complex Atari 100K benchmark [50] in Appendix E.2. Further details on the experimental setup can be found in Appendix C.

**Baselines.** To evaluate the effectiveness of our proposed CsDreamer framework, we compare it against several state-of-the-art model-based RL algorithms known for their strong performance in high-dimensional observation environments. Our primary baselines include Dreamer [17], TIA [10], Iso-Dream [12] and Denoised MDP [13]. Dreamer is a classic work in model-based RL but does not specifically address environments with noisy distractors. TIA targets settings with significant distractions by employing separate dynamics models for task-relevant and task-irrelevant features and utilizing masking techniques to isolate essential information. Iso-Dream incorporates separate task-relevant and task-irrelevant branches within a single world model to focus on aspects of the environment that the agent can influence. Denoised MDP decomposes observations into more fine-grained components, enhancing performance in complex environments with noisy distractors. Additionally, we compare our method with DBC [21], a model-free RL method, in the DMC environment with gray natural videos as background.

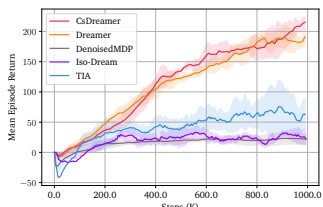 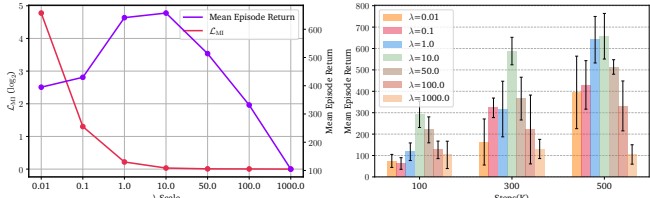

Figure 5: Performance on the autonomous driving simulator CARLA.

Figure 6: Ablation study across different $\lambda$ on *Cheetah Run* using gray driving car as background. **Left** is the relationship between mutual information upper bound and mean episode return at $500K$ timesteps. **Right** is the mean episode return over varying steps.

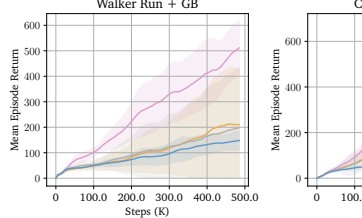 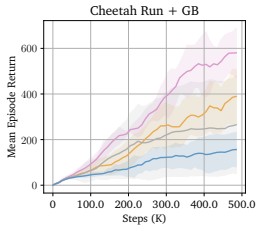 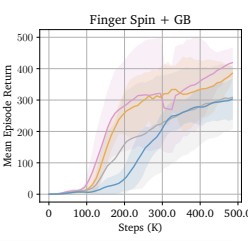 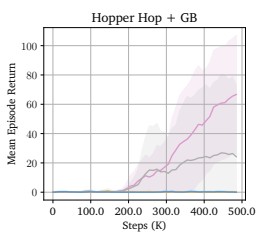

Figure 7: Ablation Performance on DMC using gray natural videos as background.

## 5.2 Performance in Visual Inputs with Complex Noisy Distractors

Figure 3 presents the performance of various methods on DMC with gray natural video backgrounds. Solid lines represent mean episode returns, while shaded areas represent the $95\%$ confidence intervals. All methods show some performance in the *Walker Run* and *Cheetah Run* tasks. While most approaches performed well on the *Finger Spin* task, the model-free method, DBC, lags behind. Only CsDreamer, TIA, and DenoisedMDP demonstrate substantial performance on the more complex *Hopper Hop* task. Notably, CsDreamer consistently outperforms all baseline methods across all four benchmark tasks. Figure 4 presents the performance on DMC with colorful backgrounds. The results also show that CsDreamer consistently outperforms all baseline methods across the four tasks. Iso-Dream exhibits the lowest performance in every task, likely due to its difficulty in managing complex and colorful background distraction. In the *Cheetah Run* and *Finger Spin* tasks, TIA shows competitive performance, approaching that of CsDreamer. In addition, Dreamer also delivers promising results within this benchmark, which may be attributed to the characteristics of the dataset used in Distracting Control Suite [48]. Specifically, it introduces background variations with multiple non-continuous images, making it easier for Dreamer to capture and disregard background-related features compared to scenarios using original videos as backgrounds, as shown in Figure 3.

We also evaluate these algorithms on the CARLA simulator, a more realistic benchmark for autonomous driving. The results are shown in Figure 5. All the methods exhibit similar performance at the first 200K timesteps. Although the interference-handling baselines such as TIA and Denoised-MDP perform well in the DMC environment with noisy background distractors, they fail to achieve comparable performance in the CARLA environment. This discrepancy arises from CARLA's substantial complexity, where these methods struggle to effectively distinguish task-relevant information from irrelevant distractions in first-person views and fail to extract task-relevant features from the highly intricate and volatile input observation. In contrast, CsDreamer demonstrates superior adaptability, outperforming all baselines and highlighting its strength in addressing more realistic and complex reinforcement learning tasks. Notably, Dreamer achieves impressive performance in the CARLA environment, slightly trailing behind CsDreamer. This strong performance is attributable to Dreamer's approach of encoding all environmental information into its latent space, which ensures the preservation of critical task-relevant features essential for autonomous driving. However, the inclusion of irrelevant information may slightly hinder its efficiency and robustness compared to CsDreamer.

## 5.3 Ablation Study

**The Role of Regularization.** Figure 6 illustrates the effect of varying the coefficient $\lambda$ in Eq. (6). As $\lambda$ increases, the upper bound of mutual information progressively decreases, signaling a reduced conditional dependence between task-relevant and task-irrelevant variables. Meanwhile, the mean

episode return exhibits a non-monotonic trend, initially increasing and then decreasing. A small $\lambda$ weakens the constraint imposed by the mutual information upper bound, heightening the risk of confounding task-relevant with task-irrelevant information, as previously discussed. Conversely, a large $\lambda$ minimizes the association between these variables. The near independence of $c_t$ and $s_t$, given the observation $o_t$, results in poorer performance, thereby validating the analysis presented in Section 4.3.

**Effectiveness of Different Modules.** We conduct ablation studies to evaluate the contributions of the conditional dependence and decoupling regularization modules in CsDreamer. Specifically, we compare the performance of **CsDreamer** and **Dreamer** against:

- **CsDreamer w/o CD**: CsDreamer using the CsRSSM framework without conditional dependence for $s_t$ and $c_t$ given $o_t$ in collider structures.
- **CsDreamer w/o MI**: CsDreamer with the CsRSSM framework excluding decoupling regularization.

The implementation details can refer to Appendix D.3. As shown in Figure 7, **CsDreamer w/o CD** achieves mean episode returns comparable to or higher than those of **Dreamer**, demonstrating the effectiveness of structural decomposition. However, it underperforms compared to **CsDreamer w/o MI**, highlighting the importance of conditional dependence. Introducing decoupling regularization, **CsDreamer** performs better than **CsDreamer w/o MI**. It demonstrates that the decoupling regularization can mitigate confusion between task-relevant and task-irrelevant variables, as mentioned in Section 4.2. **CsDreamer** consistently outperforms all other models across all tasks, achieving the best performance. It validates the effectiveness of CsRSSM and the efficient utilization of task-relevant information within the behavior policy of **CsDreamer**, ultimately enhancing overall performance. Additional ablation experiments are provided in Appendix E.5.

## 5.4 Visualization of The Latent Variables

In this section, we reconstruct the visual input observation to interpret semantic information extracted by $s_t$ and $c_t$. Figure 8 presents the reconstruction visualization results for *Cheetah Run + GB*. We sample a trajectory and record both ground truth and reconstructed images at selected timesteps. The first row displays the ground truth of the trajectory. The second row shows reconstructions using the model $p_\phi(\hat{o}_t|h_t^s, s_t, h_t^c, c_t)$ from CsRSSM, which concatenates features from both posterior $s_t$ and $c_t$. To isolate the semantic information captured by $c_t$, we set all feature dimensions of $s_t$ to zero and concatenate it with posterior $c_t$ as the features for reconstruction, denoted as $p_\phi(\hat{o}_t|\mathbf{0}, \mathbf{0}, h_t^c, c_t)$ in the third row. Conversely, to focus on information from $s_t$, we set features of $c_t$ to zero and reconstruct using $p_\phi(\hat{o}_t|h_t^s, s_t, \mathbf{0}, \mathbf{0})$, shown in the fourth row. In the fifth row, we obtain the prior $\hat{s}_t$ by imagining within the task-relevant transition model $p_\phi(\hat{s}_t|h_t^s)$ using the first five posterior $s_t$ features and then reconstruct the subsequent observations accordingly.

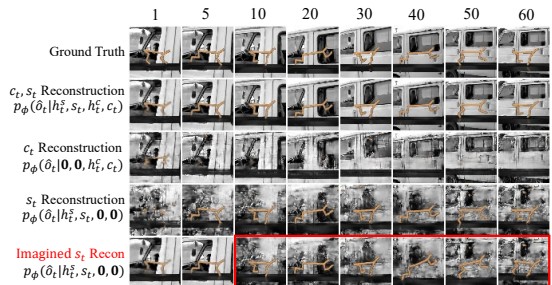

Figure 8: Reconstruction visualization of *Cheetah Run + GB*.

The second row demonstrates that combining features from $s_t$ and $c_t$ facilitates effective reconstruction. In the third row, the cheetah entity is nearly imperceptible, leaving only background automobile details. Conversely, the fourth row exhibits a highly blurred background while the cheetah remains clear. This indicates that $c_t$ effectively captures task-irrelevant information, whereas $s_t$ captures task-relevant information. In the fifth row, the cheetah in the first 30 frames closely matches the ground truth, and the background is significantly blurred. These observations suggest that performing imagination in $S$ space allows the model to focus on task-relevant dynamics. These results illustrate why CsDreamer achieves outstanding performance. Additional visualization experiments are provided in Appendix E.8.

## 6 Conclusion and Future Work

In this work, we adopt a generative modeling perspective and exploit the conditional dependence between task-relevant and task-irrelevant latent variables for both observation generation and environment dynamics in scenarios with complex noisy distractors. By effectively separating task-relevant features from irrelevant background interference, CsDreamer enhances the agent's decision-making

capability. Our experiments confirm its superior performance in noisy environments. We also acknowledge the limitations, including the assumption of Markovian, action-independent noise and a simplified binary partition of latent variables, which may not capture the complexity of real-world scenarios. The model's capability carries some societal implications. While it can improve safety in applications like autonomous navigation and robotics, it also presents some risks. These include overreliance, where a critical safety signal could be erroneously ignored, and dual-use potential, where the technology could be repurposed for invasive surveillance. Our future work will address these challenges on two fronts. Technically, we will focus on modeling more complex non-Markovian dynamics and exploring richer, more structured latent representations. Concurrently, we will prioritize research into enhancing model transparency to mitigate overreliance and will investigate technical safeguards, such as built-in privacy-preserving mechanisms, to deter misuse. We believe this dual focus on technical advancement and ethical considerations is essential for the responsible development of robust autonomous agents.

## Acknowledgments and Disclosure of Funding

We would like to thank Yucen Wang and the anonymous reviewers for their helpful discussions and support. This work was supported by the National Science Foundation of China (62476123) and the Young Scientists Fund of the National Natural Science Foundation of China (PhD Candidate) (624B200197).

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

# A  Detailed Derivations

## A.1  Derivation of ELBO Loss in CsRSSM

We use $o_{1:T}$, $a_{1:T}$, $r_{1:T}$ and $x_{1:T}$ for observation sequence, action sequence, reward sequence and episode continuation flag sequence. Similarly, we also use $s_{1:T}$ and $c_{1:T}$ for task-relevant and task-irrelevant latent variables. After introducing the variational posterior $q(s_{1:T}, c_{1:T}|o_{1:T}, a_{1:T}) = \prod_t q(c_t|o_{\leq t}, a_{<t})q(s_t|c_t, o_{\leq t}, a_{<t})$, we can obtain the variational bound for the dynamics models $p((o_{1:T}, r_{1:T}, x_{1:T}), s_{1:T}, c_{1:T}|a_{1:T}) = \prod_t p(s_t|s_{t-1}, a_{t-1})p(c_t|c_{t-1})p(o_t|s_t, c_t)p(r_t|s_t)p(x_t|s_t)$ in CsRSSM using Jensen's inequality

$$
\begin{aligned}
&\ln p((o_{1:T}, r_{1:T}, x_{1:T})|a_{1:T}) \\
=&\ln \left[ \mathbb{E}_{q(s_{1:T}, c_{1:T}|o_{1:T}, a_{1:T})} \left[ \frac{p((o_{1:T}, r_{1:T}, x_{1:T}), s_{1:T}, c_{1:T}|a_{1:T})}{q(s_{1:T}, c_{1:T}|o_{1:T}, a_{1:T})} \right] \right] \\
\geq&\mathbb{E}_{q(s_{1:T}, c_{1:T}|o_{1:T}, a_{1:T})} \left[ \ln \frac{p((o_{1:T}, r_{1:T}, x_{1:T}), s_{1:T}, c_{1:T}|a_{1:T})}{q(s_{1:T}, c_{1:T}|o_{1:T}, a_{1:T})} \right] \\
=&\mathbb{E}_{q(s_{1:T}, c_{1:T}|o_{1:T}, a_{1:T})} \left[ \ln \frac{\prod_t p(s_t|s_{t-1}, a_{t-1})p(c_t|c_{t-1})p(o_t|s_t, c_t)p(r_t|s_t)p(x_t|s_t)}{\prod_t q(c_t|o_{\leq t}, a_{<t})q(s_t|c_t, o_{\leq t}, a_{<t})} \right] \\
=&\mathbb{E}_{q(s_{1:T}, c_{1:T}|o_{1:T}, a_{1:T})} \left[ \sum_{t=1}^{T} [\ln p(s_t|s_{t-1}, a_{t-1}) + \ln p(c_t|c_{t-1}) + \ln p(o_t|s_t, c_t) + \ln p(r_t|s_t) + \ln p(x_t|s_t) \right. \\
&\left. \quad - \ln q(c_t|o_{\leq t}, a_{<t}) - \ln q(s_t|c_t, o_{\leq t}, a_{<t})] \right] \qquad (7) \\
=&\sum_{t=1}^{T} \left[ \int \prod_{t'=1}^{T} q(c_{t'}|o_{\leq t'}, a_{<t'})q(s_{t'}|c_{t'}, o_{\leq t'}, a_{<t'}) [\ln p(o_t|s_t, c_t) + \ln p(r_t|s_t) + \ln p(x_t|s_t) \right. \\
&\left. \quad + (\ln p(s_t|s_{t-1}, a_{t-1}) - \ln q(s_t|c_t, o_{\leq t}, a_{<t})) + (\ln p(c_t|c_{t-1}) - \ln q(c_t|o_{\leq t}, a_{<t}))] \, \mathrm{d}s_{1:T}\mathrm{d}c_{1:T} \right] \\
=&\sum_{t=1}^{T} \left[ \mathbb{E}_{s_t, c_t}[\ln p(o_t|s_t, c_t)] + \mathbb{E}_{s_t}[\ln p(r_t|s_t)] + \mathbb{E}_{s_t}[\ln p(x_t|s_t)] - \mathbb{E}_{c_{t-1}}[\mathrm{KL}[q(c_t|o_{\leq t}, a_{<t})\|p(c_t|c_{t-1})]] \right. \\
&\left. \quad - \mathbb{E}_{c_t, s_{t-1}}[\mathrm{KL}[q(s_t|c_t, o_{\leq t}, a_{<t})\|p(s_t|s_{t-1}, a_{t-1})]] \right].
\end{aligned}
$$

By applying the history models in Eq. (3), we can ultimately obtain the $\mathcal{L}_{\mathrm{ELBO}}$ in Eq. (4).

## A.2  Derivation of the upper bound for conditional mutual information

By incorporating the variational approximate distribution $q(s_t|h_t^c, h_t^s, o_t)$, the original conditional mutual information is given by:

$$
\begin{aligned}
I(s_t; c_t|h_t^c, h_t^s, o_t) =&\mathbb{E}_{o_t, s_t, c_t} \left[ \ln \frac{p(s_t, c_t|h_t^c, h_t^s, o_t)}{p(s_t|h_t^c, h_t^s, o_t)p(c_t|h_t^c, h_t^s, o_t)} \right] \\
=&\mathbb{E}_{o_t, s_t, c_t} \left[ \ln \frac{p(s_t|c_t, h_t^c, h_t^s, o_t)q(s_t|h_t^c, h_t^s, o_t)}{p(s_t|h_t^c, h_t^s, o_t)q(s_t|h_t^c, h_t^s, o_t)} \right] \\
=&\mathbb{E}_{o_t, s_t, c_t} \left[ \ln \frac{p(s_t|c_t, h_t^c, h_t^s, o_t)}{q(s_t|h_t^c, h_t^s, o_t)} \right] - \mathbb{E}_{o_t, s_t, c_t} \left[ \ln \frac{p(s_t|h_t^c, h_t^s, o_t)}{q(s_t|h_t^c, h_t^s, o_t)} \right] \\
=&\mathbb{E}_{o_t, s_t, c_t} \left[ \ln \frac{p(s_t|c_t, h_t^c, h_t^s, o_t)}{q(s_t|h_t^c, h_t^s, o_t)} \right] - \mathbb{E}_{o_t, s_t} \left[ \ln \frac{p(s_t|h_t^c, h_t^s, o_t)}{q(s_t|h_t^c, h_t^s, o_t)} \right] \\
=&\mathbb{E}_{o_t, c_t}[\mathrm{KL}[p(s_t|c_t, h_t^c, h_t^s, o_t)\|q(s_t|h_t^c, h_t^s, o_t)]] - \mathbb{E}_{o_t}[\mathrm{KL}[p(s_t|h_t^c, h_t^s, o_t)\|q(s_t|h_t^c, h_t^s, o_t)]] \\
\leq&\mathbb{E}_{o_t, c_t}[\mathrm{KL}[p(s_t|c_t, h_t^c, h_t^s, o_t)\|q(s_t|h_t^c, h_t^s, o_t)]].
\end{aligned} \qquad (8)
$$

Then we can obtain the upper bound regularization loss in Eq. (5).

## B Pseudo Code

The whole algorithm is shown in Algorithm 1. For brevity, we omit the episode continuation flag $x_t$.

---

**Algorithm 1** CsDreamer

---

**Initialize:** Dataset $\mathcal{D}$ collected by random policy, the policy parameters $\theta$, the critic parameters $\psi$, the variational estimator parameters $\xi$ and the parameters $\phi$ in the CsRSSM world model
**for** training step $t_1 = 1...T_1$ **do**
    **for** update step $t_2 = 1...T_2$ **do**
        // `Dynamics learning`
        Sample minibatch $\{(o_t, a_t, r_t)\}_{t=k}^{k+L} \sim \mathcal{D}$
        Compute loss according to Eq. (6)
        Update CsRSSM world model parameters $\phi$ and the variational estimator parameters $\xi$
        // `Behavior learning`
        Infer the task-irrelevant information $c_t \sim q_\phi(c_t|h_t^c, h_t^s, o_t)$
        Infer the task-relevant information $s_t \sim q_\phi(s_t|h_t^c, h_t^s, c_t, o_t)$
        Imagine trajectories $\{(s_t, a_t, r_t)\}_{t=k}^{k+H}$ using $p_\phi(\hat{s}_t|h_t^s), p_\phi(\hat{r}_t|h_t^s, s_t)$ and the policy $\pi$
        Update the policy parameters $\theta$ and the critic parameters $\psi$ using the imagined trajectories
    **end for**
    **for** rollout step $t_3 = 1...T_3$ **do**
        Infer the task-irrelevant information $c_t \sim q_\phi(c_t|h_t^c, h_t^s, o_t)$
        Infer the task-relevant information $s_t \sim q_\phi(s_t|h_t^c, h_t^s, c_t, o_t)$
        Sample action from exploration policy $a_t \sim \pi_{\exp}(a_t|h_t^s, s_t)$
        $r_t, o_{t+1} \leftarrow$ `env.step`$(a_t)$
    **end for**
    Add experience to dataset $\mathcal{D} \leftarrow \mathcal{D} \cup \{(o_t, a_t, r_t)\}$
**end for**

---

## C Details about the Benchmark

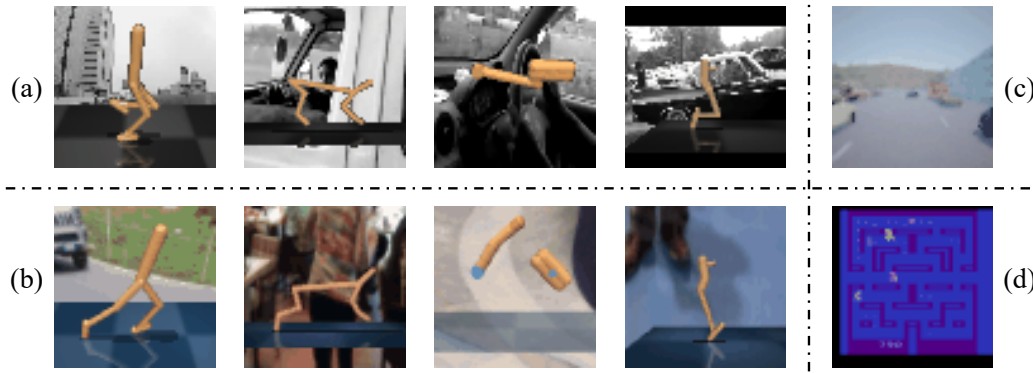

Figure 9: Example observations of the benchmarks. (**a**) The observations in DMC with gray videos in the background and the four tasks from left to right are *Walker Run*, *Cheetah Run*, *Finger Spin* and *Hopper Hop*. (**b**) The observations in DMC with colourful videos in the background and the four tasks from left to right are *Walker Run*, *Cheetah Run*, *Finger Spin* and *Hopper Hop*. (**c**) The observation in CARLA simulator. (**d**) The observation in the game *Alien*, which is one of the 26 games in the Atari 100K benchmark.

**DMC with Complex Backgrounds.** We introduce the natural video background as the noisy distractor for the widely used robotic locomotion benchmark, DeepMind Control Suite [45]. We select four tasks for evaluation. The *Walker Run* task assesses the stability and coordination of bipedal locomotion, *Cheetah Run* examines high-speed movement and agility, *Finger Spin* evaluates

precise motor control and object manipulation capabilities, and *Hopper Hop* tests dynamic balance and energy management in single-leg hopping. We choose *Walker Run* and *Hopper Hop* instead of *Walker Walk* and *Hopper Stand* for more challenging evaluation. For each task, we adopt two types of noisy distractors. We first introduce the natural video from the Kinetics dataset [46]. We only use the videos in 'driving car' class and set it to gray as that in [21, 10] (Figure 9 (**a**)). Then we utilize the colourful videos of the DAVIS 2017 dataset [47] as that in [48]. We utilize all the 90 train-val videos and adopt the dynamic setting, where the video plays forwards or backwards until the last or first frame is reached at which point the playing direction is reversed, thereby the background motion is always smooth and without 'cuts' (Figure 9 (**b**)). The height and the width of the input observation image are $64 \times 64$.

**CARLA.**    To evaluate on a more real-world control system, we use the CARLA simulator [49] to conduct photo-realistic visual observations. The agent's goal is to drive as far as possible along CARLA's Town04's highway in 1000 timesteps in a first-person way without colliding with other moving vehicles or barriers as the setting in DBC [21]. To increase the task difficulty, we make two modifications as that in Iso-Dream setting [12]. We use one camera which obtains images of $64 \times 64$ pixels instead of five to limit the field of view and we include 30 other moving vehicles or obstacles instead of 20 to increase the likelihood of collisions. The example observation is shown in Figure 9 (**c**).

**Atari 100K Benchmark.**    The Atari 100K benchmark [50] comprises 26 distinct Atari games. Within this benchmark, an agent is permitted 100K interaction steps for each game environment. Due to a frameskip setting of 4, this translates to 400K frames. This interaction volume is roughly equivalent to about two hours of real-time gameplay. The game environments within this benchmark are notably complex, making it a practical testbed for evaluating the algorithms' robustness and data efficiency. We keep all implementation details the same as Dreamer [17], and the example observation is shown in Figure 9 (**d**).

# D    Implementation Details

## D.1    Base Method

CsDreamer is implemented based on the classic model-based reinforcement learning method, Dreamer [18, 19, 17]. Given that Dreamer has multiple versions, we first evaluate DreamerV2 [19] and DreamerV3 [17] [2] on DMC using gray natural videos as background.

The performance results are in Figure 10, both DreamerV2 and DreamerV3 demonstrate overall similar performance across the four tasks. Specifically, DreamerV3 shows a slight advantage in the *Finger Spin + GB* tasks, while DreamerV2 performs marginally better in the *Walker Run + GB* and *Hopper Hop + GB* tasks. They have comparable performance in the *Cheetah Run + GB* task. Overall, the performance differences between the two versions are minimal. The minimal performance differences between DreamerV2 and DreamerV3 can likely be attributed to several factors. Firstly, both models share a significant degree of architectural and methodological overlap. Secondly, some of the improvements in DreamerV3 are primarily designed to enhance adaptability across various domains and tasks. However, these enhancements may compromise the model's fundamental capabilities. This trade-off means that while DreamerV3 can perform effectively in a broader range of scenarios, it might not significantly outperform DreamerV2, resulting in similar performance across the evaluated tasks. Considering that DreamerV3 has a much smaller world model loss (on the order of tens) than DreamerV2 (over ten thousand) and in our experiments $\mathcal{L}_{\text{MI}}$ is typically within single digits, we use DreamerV3 as the base method so that $\mathcal{L}_{\text{CsRSSM}}$ and $\mathcal{L}_{\text{MI}}$ have a similar order of magnitude to avoid problems. Unless otherwise specified, in the experiments, **Dreamer** refers to DreamerV3, and **CsDreamer** is based on DreamerV3.

---

[2]In this paper, we utilize the 'S' size model for DreamerV3 in https://arxiv.org/pdf/2301.04104v1.

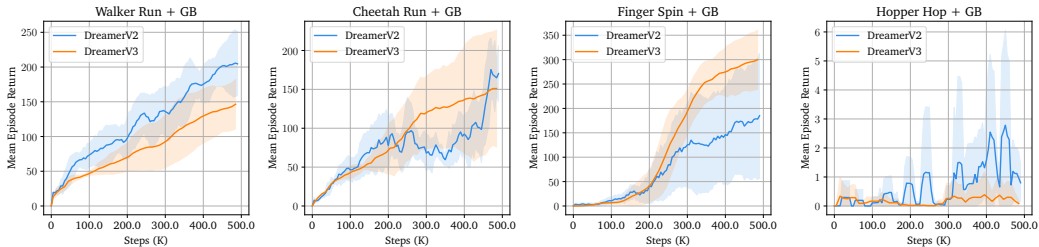

Figure 10: Performance of DreamerV2 and DreamerV3 on DMC using the gray natural videos as background.

## D.2 Key Component Implementation

The RSSM world model in Dreamer implements the KL divergence as two separate components in different stop-gradient operator places and loss scales [17]. Similarly, the two KL divergences in Eq. (4) are each composed of two distinct loss terms. Specifically, we have

$$
\begin{aligned}
\mathbb{E}[\mathrm{KL}[q_\phi(c_t|h_t^c, h_t^s, o_t)\|p_\phi(c_t|h_t^c)]] &= \beta_{\mathrm{dyn}}^c \mathcal{L}_{\mathrm{dyn}}^c(\phi) + \beta_{\mathrm{rep}}^c \mathcal{L}_{\mathrm{rep}}^c(\phi) \\
\mathbb{E}[\mathrm{KL}\,[q_\phi(s_t|h_t^c, h_t^s, c_t, o_t)\|p_\phi(s_t|h_t^s)]] &= \beta_{\mathrm{dyn}}^s \mathcal{L}_{\mathrm{dyn}}^s(\phi) + \beta_{\mathrm{rep}}^s \mathcal{L}_{\mathrm{rep}}^s(\phi),
\end{aligned}
\tag{9}
$$

where

$$
\begin{aligned}
\mathcal{L}_{\mathrm{dyn}}^c(\phi) &= \max\left(1, \mathrm{KL}\left[\mathrm{sg}\left(q_\phi(c_t|h_t^c, h_t^s, o_t)\right)\|p_\phi(c_t|h_t^c)\right]\right) \\
\mathcal{L}_{\mathrm{rep}}^c(\phi) &= \max\left(1, \mathrm{KL}\left[q_\phi(c_t|h_t^c, h_t^s, o_t)\|\mathrm{sg}\left(p_\phi(c_t|h_t^c)\right)\right]\right) \\
\mathcal{L}_{\mathrm{dyn}}^s(\phi) &= \max\left(1, \mathrm{KL}\left[\mathrm{sg}\left(q_\phi(s_t|h_t^c, h_t^s, c_t, o_t)\right)\|p_\phi(s_t|h_t^s)\right]\right) \\
\mathcal{L}_{\mathrm{rep}}^s(\phi) &= \max\left(1, \mathrm{KL}\left[q_\phi(s_t|h_t^c, h_t^s, c_t, o_t)\|\mathrm{sg}\left(p_\phi(s_t|h_t^s)\right)\right]\right),
\end{aligned}
\tag{10}
$$

and $\mathrm{sg}(\cdot)$ denotes the stop-gradient operator. In the CsRSSM world model, the task-relevant component utilizes the same network architecture as Dreamer. The primary distinction arises during the computation of the observation posterior, where both the observation $o_t$ and the inferred latent variable $c_t$ are concatenated and provided as input, rather than using only the observation $o_t$ as in Dreamer. Additionally, when reconstructing the observation, the model concatenates the inferred $c_t$ with the variable $s_t$ to reconstruct $o_t$ jointly. The task-irrelevant component employs a network structure similar to that of Dreamer but with some modifications. It does not take actions as input, ensuring that it remains unaffected by action-related information. Moreover, it does not predict task-relevant variables such as rewards, thereby focusing solely on modeling the noise distractors independent of the task-relevant objectives.

For the regularization loss $\mathcal{L}_{\mathrm{MI}}$, we use a variational estimator $q_\xi$ for the variational approximate distribution $q(s_t|h_t^c, h_t^s, o_t)$. The variational estimator takes the history $(h_t^c, h_t^s)$, and the embedding of $o_t$ as the input. The gradients from these variables will be blocked from backpropagation during the training of the variational estimator, i.e., $q_\xi(s_t|\mathrm{sg}(h_t^c, h_t^s, o_t))$. This ensures that the learning of the variational estimator does not influence the parameters of the preceding models. Then, we utilize the part network of $c_t$ to get the embedding of $(h_t^c, h_t^s, o_t)$, and we use the embedding to get the variational approximate distribution after a linear layer.

## D.3 Module Ablation Implementation

Here, we detail the implementation of the two ablation methods described in Section 5.3. **CsDreamer w/o MI** is implemented by simply removing the decoupling regularization from **CsDreamer**. **CsDreamer w/o CD** adopts the CsRSSM framework with some modifications. Specifically, since **CsDreamer w/o CD** does not account for the conditional dependence between $s_t$ and $c_t$ given $o_t$ in collider structures, we adjust its encoders: the task-irrelevant encoder is defined as $q_\phi(c_t|h_t^c, o_t)$ and the task-relevant encoder is defined as $q_\phi(s_t|h_t^s, o_t)$. The world model is then trained using $\mathcal{L}_{\mathrm{ELBO}}$. Because **CsDreamer w/o CD** ignores all conditional dependence, the $\mathcal{L}_{\mathrm{MI}}$ term is omitted.

### D.4 Hyperparameters and Time Cost

Table 1 presents the primary hyperparameters of CsDreamer. Since the behavior policy relies solely on the feature of $s_t$, and the hyperparameters for $s_t$ in CsRSSM closely resemble those of the latent variables in RSSM of DreamerV3, we adopt the same hyperparameters for the behavior policy as in DreamerV3. The experiments are mainly conducted on NVIDIA RTX 4090 GPUs. With each GPU, we are able to train each environment at a rate of approximately 24K timesteps per hour.

Table 1: Hyperparameters for CsDreamer

| Hyperparameter | Value |
|---|---|
| Action Repeat | 4 for CARLA and Atari, and 2 for others |
| $\lambda$ | 10.0 for Hopper Hop+GB, 0.2 for CARLA and Atari, and 1.0 for others |
| $\beta_{\mathrm{dyn}}^{s}$ | 0.5 |
| $\beta_{\mathrm{rep}}^{s}$ | 0.1 |
| $\beta_{\mathrm{dyn}}^{c}$ | 0.5 |
| $\beta_{\mathrm{rep}}^{c}$ | 0.1 |
| Discrete latent dimensions of $s_t$ | 32 |
| Discrete latent classes of $s_t$ | 32 |
| Discrete latent dimensions of $c_t$ | 16 for CARLA, 8 for Atari, and 32 for others |
| Discrete latent classes of $c_t$ | 16 for CARLA, 8 for Atari, and 32 for others |
| GRU recurrent units of $s_t$ | 512 |
| GRU recurrent units of $c_t$ | 256 for CARLA, 128 for Atari, and 512 for others |
| Dense hidden units of $s_t$ | 512 |
| Dense hidden units of $c_t$ | 512 |
| MLP layers | 2 |

## E  Additional Experiment Results

### E.1  Performance Scores

In Table 2, we summarize the final mean episode return and their corresponding standard deviations for various model-based RL methods across multiple environments in the main text, evaluated using four distinct random seeds, each associated with 10 evaluation episodes. The results consistently show that CsDreamer outperforms the other methods in the majority of environments, thereby demonstrating the superior effectiveness of our proposed approach.

Table 2: Final performance across model-based RL methods in different environments.

| Environment | CsDreamer (Ours) | Dreamer | TIA | Denoised MDP | Iso-Dream |
|---|---|---|---|---|---|
| Walker Run + GB | $\mathbf{533 \pm 98}$ | $162 \pm 32$ | $293 \pm 129$ | $117 \pm 81$ | $131 \pm 23$ |
| Cheetah Run + GB | $\mathbf{547 \pm 159}$ | $171 \pm 84$ | $432 \pm 172$ | $215 \pm 191$ | $109 \pm 12$ |
| Finger Spin + GB | $408 \pm 45$ | $287 \pm 54$ | $123 \pm 201$ | $331 \pm 146$ | $\mathbf{415 \pm 118}$ |
| Hopper Hop + GB | $\mathbf{75 \pm 58}$ | $0 \pm 0$ | $\mathbf{50 \pm 48}$ | $13 \pm 15$ | $0 \pm 0$ |
| Walker Run + CB | $\mathbf{678 \pm 74}$ | $474 \pm 71$ | $588 \pm 135$ | $85 \pm 18$ | $211 \pm 95$ |
| Cheetah Run + CB | $\mathbf{821 \pm 58}$ | $549 \pm 126$ | $758 \pm 80$ | $392 \pm 195$ | $6 \pm 4$ |
| Finger Spin + CB | $\mathbf{576 \pm 241}$ | $466 \pm 130$ | $490 \pm 136$ | $460 \pm 123$ | $4 \pm 8$ |
| Hopper Hop + CB | $\mathbf{70 \pm 52}$ | $23 \pm 28$ | $42 \pm 40$ | $21 \pm 19$ | $0 \pm 0$ |
| CARLA | $\mathbf{235 \pm 118}$ | $202 \pm 144$ | $44 \pm 119$ | $21 \pm 28$ | $24 \pm 25$ |

### E.2  Performance on Atari 100K Benchmark

In order to assess the denoising capabilities of CsDreamer under highly complex visual inputs, we conduct experiments on the Atari 100K benchmark, and the results are presented in Figure 11 and Table 3. For each game, we use at least five seeds. During our experiments, we encounter a random seed that produces performance approximately 20 times higher than the current baseline, which we

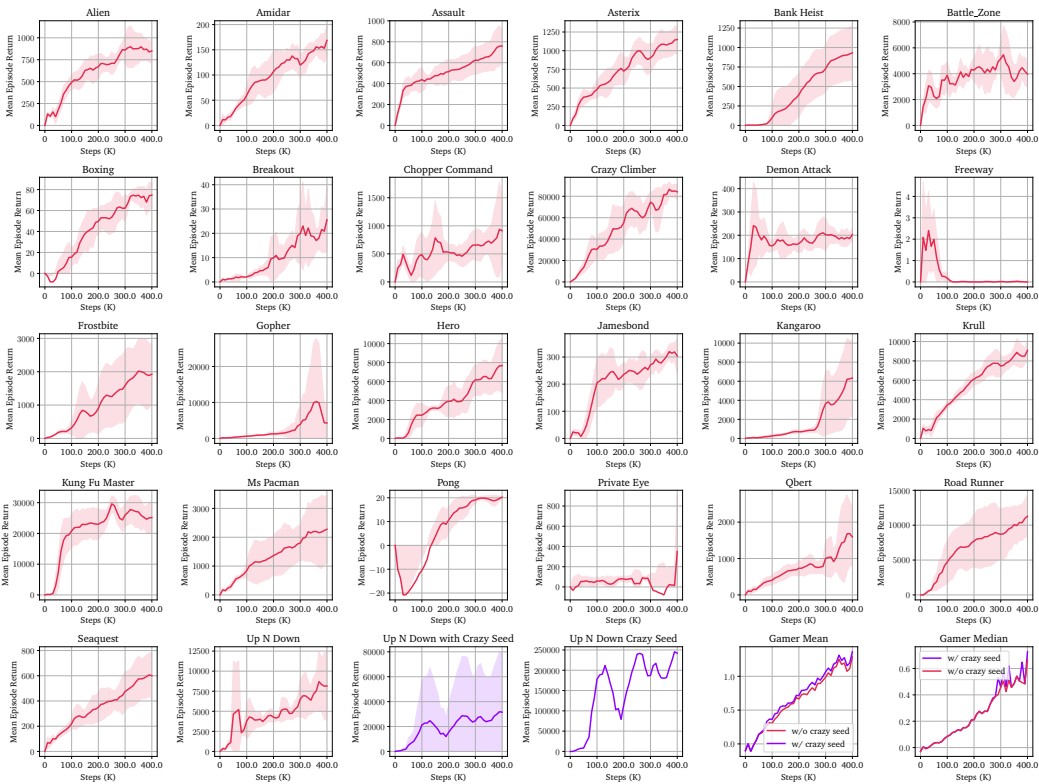

Figure 11: CsDreamer performance on Atari 100K Benchmark. In the *Up N Down* environment, we encounter a random seed that yields performance approximately 20 times higher than the current baseline, which we call the 'crazy seed'. We use 10 seeds in this environment to ensure a more accurate evaluation. The *Up N Down* results reflect the performance of the 9 seeds, excluding the crazy seed, while *Up N Down with Crazy Seed* presents the outcomes incorporating all 10 seeds.

call the 'crazy seed'. To facilitate a more robust and accurate evaluation, we employ a total of 10 seeds in this environment.

In Figure 11, the results labeled *Up N Down* reflect the performance obtained from nine seeds (excluding the crazy seed), whereas *Up N Down with Crazy Seed* encompasses outcomes derived from all 10 seeds. In Table 3, values within parentheses indicate results achieved using the crazy seed, while results outside the parentheses correspond to evaluations without its inclusion. The Dreamer results in Table 3 are sourced directly from the official DreamerV3 paper [17] [3]. The experimental outcomes clearly indicate that CsDreamer outperforms the baseline approach. This improvement demonstrates that leveraging conditional dependencies enables CsDreamer to effectively denoise complex visual inputs, thereby enhancing overall task performance.

## E.3 Performance on Standard DMC

We conduct experiments using the standard DeepMind Control (DMC) suite to evaluate performance on clean benchmarks. Figure 12 illustrates performance across four tasks. As depicted, CsDreamer generally achieves performance on par with the original Dreamer algorithm across all four evaluated tasks. These results indicate that the CsDreamer algorithm effectively handles noisy scenarios and maintains strong performance in noise-free environments.

---

[3]In this paper, we utilize the data in https://arxiv.org/pdf/2301.04104v1.

Table 3: CsDreamer's performance on the Atari 100K Benchmark. Values in parentheses indicate the use of the 'crazy seed'.

| Game | Random | Human | Dreamer (official) | CsDreamer |
|---|---|---|---|---|
| **Alien** | 228 | 7128 | **959** | $888 \pm 234$ |
| **Amidar** | 6 | 1720 | 139 | $\mathbf{184 \pm 43}$ |
| **Assault** | 222 | 742 | 706 | $\mathbf{748 \pm 262}$ |
| **Asterix** | 210 | 8503 | 932 | $\mathbf{1114 \pm 266}$ |
| **Bank Heist** | 14 | 753 | 649 | $\mathbf{946 \pm 465}$ |
| **Battle Zone** | 2360 | 37188 | **12250** | $3960 \pm 2537$ |
| **Boxing** | 0 | 12 | **78** | $\mathbf{80 \pm 13}$ |
| **Breakout** | 2 | 30 | **31** | $\mathbf{31 \pm 51}$ |
| **Chopper Com.** | 811 | 7388 | 420 | $\mathbf{864 \pm 480}$ |
| **Crazy Climber** | 10780 | 35829 | **97190** | $90196 \pm 22281$ |
| **Demon Attack** | 152 | 1971 | **303** | $207 \pm 124$ |
| **Freeway** | 0 | 30 | 0 | $0 \pm 0$ |
| **Frostbite** | 65 | 4335 | 909 | $\mathbf{2177 \pm 995}$ |
| **Gopher** | 258 | 2412 | 3730 | $\mathbf{7771 \pm 18563}$ |
| **Hero** | 1027 | 30826 | **11161** | $8124 \pm 3345$ |
| **James Bond** | 29 | 303 | **445** | $292 \pm 147$ |
| **Kangaroo** | 52 | 3035 | 4098 | $\mathbf{6590 \pm 4797}$ |
| **Krull** | 1598 | 2666 | 7782 | $\mathbf{9636 \pm 3531}$ |
| **Kung Fu Master** | 258 | 22736 | 21420 | $\mathbf{24847 \pm 8928}$ |
| **Ms Pacman** | 307 | 6952 | 1327 | $\mathbf{2170 \pm 1306}$ |
| **Pong** | $-21$ | 15 | 18 | $\mathbf{20 \pm 1}$ |
| **Private Eye** | 25 | 69571 | 882 | $\mathbf{977 \pm 1813}$ |
| **Qbert** | 164 | 13455 | **3405** | $1050 \pm 692$ |
| **Road Runner** | 12 | 7845 | **15565** | $11980 \pm 3946$ |
| **Seaquest** | 68 | 42055 | **618** | $578 \pm 252$ |
| **Up N Down** | 533 | 11693 | 7667 | $\mathbf{9800 \pm 13122}(\mathbf{32050 \pm 69863})$ |
| **Human Mean** | 0% | 100% | 112% | $\mathbf{129\%}(\mathbf{136\%})$ |
| **Human Median** | 0% | 100% | 49% | $\mathbf{66\%}(\mathbf{73\%})$ |

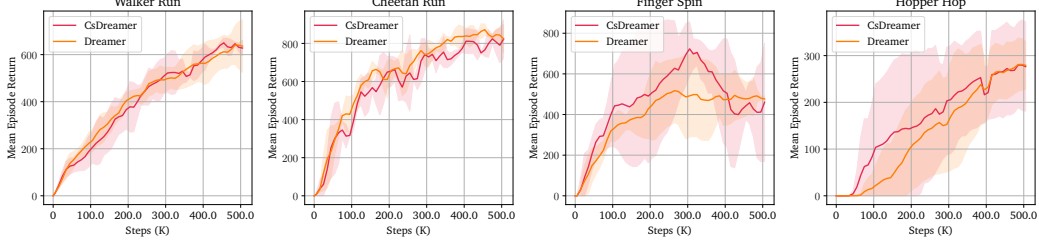

Figure 12: Performance on Standard DMC.

## E.4 Performance on DMC + GB Using Train-Eval Split

In the experiments above on **DMC + GB**, we follow previous work by training and testing using the same dataset as background. Specifically, we utilize 16 car videos beginning with the letter 'A' from the Kinetics-400 dataset, mentioned in TIA [10]. We download the complete' driving car' class using the GitHub repository[4] to ascertain whether CsDreamer's robustness is a result of merely memorizing background information or if it genuinely possesses an intrinsic denoising capability. Due to the reasons mentioned in the repository, we succeed in obtaining 641 videos. Then we alphabetically split the dataset into a training dataset (512 videos) and an evaluation dataset (129 videos) using an 80 : 20 ratio. We choose the best-performing baseline TIA in **DMC + GB** as the baseline, and the experimental results are presented in Figure 13. Compared to the results in **DMC + GB** above, the outcomes for *Walker Run*, *Cheetah Run*, and *Finger Spin* are slightly inferior, while the outcome for *Hopper Hop* is somewhat superior. The results depicted in the figure demonstrate that CsDreamer outperforms TIA across all tasks. This confirms that our approach effectively denoises rather than merely memorizing backgrounds.

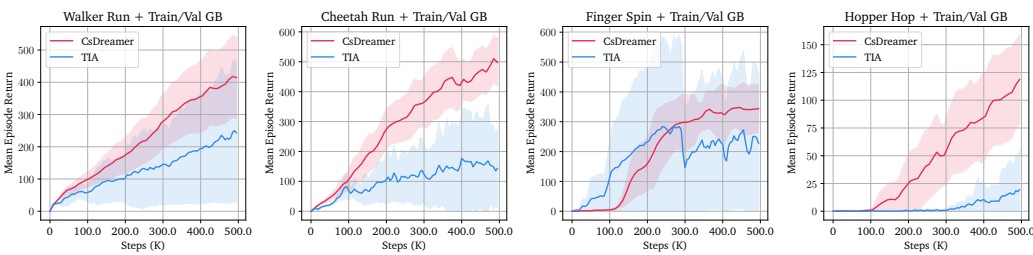

Figure 13: Performance on the train-eval dataset, with 641 driving car videos in total, 512 videos for the training dataset, and 129 videos for the evaluation dataset.

## E.5 Additional Ablation Results

We conduct more module ablation studies on the DMC benchmark using colorful natural videos as the background. Similar to that in Section 5.3, we compare the performance of the baseline **Dreamer**, **CsDreamer w/o MI** (CsDreamer with CsRSSM framework without the conditional mutual information-based regularization), **CsDreamer w/o CD** (CsDreamer with CsRSSM framework without the conditional dependence) and **CsDreamer**. Shown in Figure 14, the results are consistent with those in Section 5.3. The comparison between **Dreamer** and **CsDreamer w/o MI** reveals that the CsRSSM can significantly boost performance in visual input with complex distractors by utilizing the conditional dependence. By introducing the decoupling regularization, **CsDreamer** consistently outperforms baselines on all tasks, achieving the highest performance. This ablation study also demonstrates that CsRSSM with the decoupling information regularization significantly enhances learning efficiency and task performance.

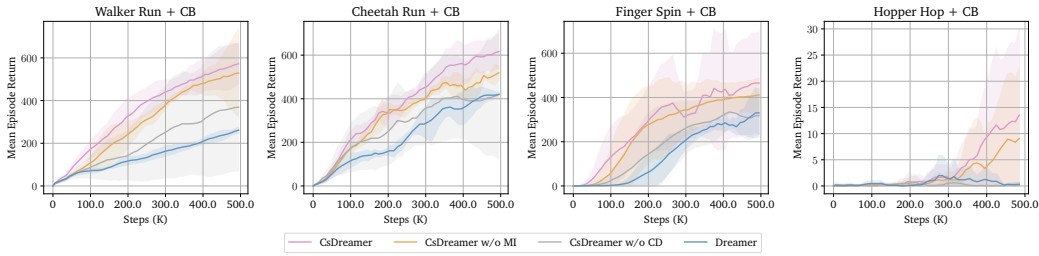

Figure 14: Ablation Performance on DMC using colorful background.

We also examine the role of regularization in the *Walker Run + GB* as in Section 5.3. As illustrated in Figure 15, the results echo those from Section 5.3: as the regularization parameter $\lambda$ increases,

---

[4]We use the Github repository in https://github.com/Showmax/kinetics-downloader.

the upper bound of mutual information progressively decreases, while the mean return follows a non-monotonic trend—initially rising and then falling. All these experiments validate the analysis presented in Section 4.3.

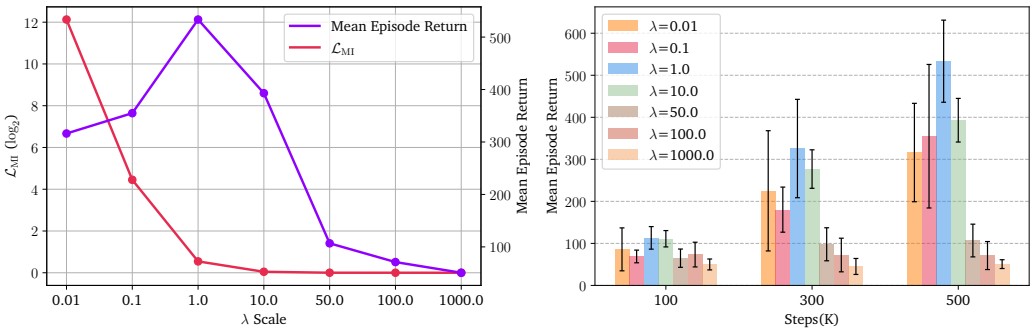

Figure 15: Ablation study across different $\lambda$ on *Walker Run* using gray driving car as background.

## E.6   Correlation analysis

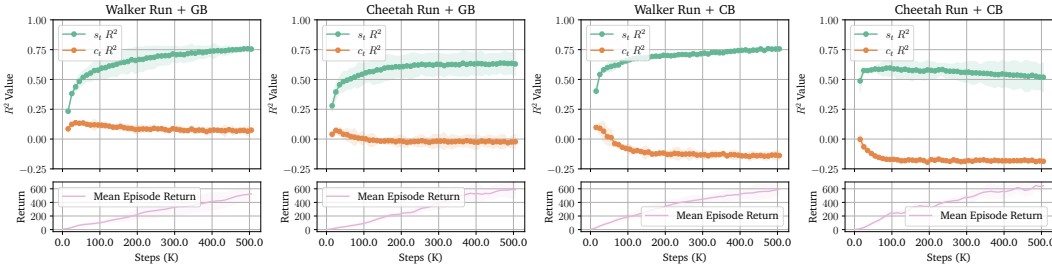

Figure 16: Coefficient of determination ($R^2$) on DMC tasks with background distractors.

In this section, we assess the coefficient of determination ($R^2$) within our framework. We find that the distribution of the test data significantly influences the $R^2$ values. Specifically, evaluating $R^2$ using the data sampled from the current policy may result in artificially high values. To obtain a more reliable assessment, we use the sampled data from the final phase of a prior training session as our dataset because, at this stage, the policy has sufficiently learned to perform the task effectively, leading to the observation continuously undergoing noticeable changes. In detail, we utilize the last 20 sampled episodes from a previous training process, with each episode comprising 1,000 timesteps (derived from 500 data points with an action repeat factor of 2). Each data point has one observation and one corresponding agent state. We utilize the observation in these episodes to yield the features for variables $s_t$ and $c_t$. We use the feature of $s_t$ and the feature of $c_t$ as the independent variables separately to predict the agent's real state. To mitigate potential interference from newly introduced training parameters, we utilize a parameter-free k-nearest neighbors (KNN) regressor for prediction. The results are shown in Figure 16. The $R^2$ value for $s_t$ rapidly increases in the early stages and stabilizes, demonstrating a strong and consistent correlation. In contrast, the $R^2$ value for $c_t$ decreases and remains low throughout the experiments. The results highlight that $s_t$ exhibits a much stronger correlation to the real agent state than $c_t$, suggesting that $s_t$ can effectively capture the task-relevant information.

## E.7   Additional Results Using Transformer-Based Framework

To clarify whether the proposed collider-structure approach is restricted to RSSM-based designs or if it is compatible with transformer-style sequence encoders, we extended our methodology to TransDreamer [51]. TransDreamer utilizes a Transformer State-Space Model (TSSM) to capture long-term dependencies, replacing the RSSM of Dreamer. We integrate our approach into this baseline, resulting in the Collider-structure Transformer State-Space Model (CsTSSM) and the corresponding CsTransDreamer. A primary adaptation challenge stems from TransDreamer's architecture, which

is optimized for parallel training. To achieve this, TSSM employs a *Myopic Representation Model*, where the posterior inference $q(z_t|o_t)$ is independent of the deterministic history state $h_t$. CsTSSM adapts the collider framework within this constraint.

The key modifications in CsTSSM are as follows:

- **Dual Transformer Dynamics**, where we replace the single dynamic model with two independent Transformer networks. One models the task-relevant dynamics, generating $h_t^s$ from past relevant states and actions; the other models the task-irrelevant dynamics, generating $h_t^c$ solely from past task-irrelevant states.
- **Parallel Collider Inference**, where we implement the collider-structure inference $p_\phi(c_t, s_t \mid o_t) = p_\phi(c_t \mid o_t)\, p_\phi(s_t \mid c_t, o_t)$. Crucially, because this inference path does not depend on the history states, the entire sequence can be encoded in parallel across the time dimension before the Transformer dynamics are computed, preserving the efficiency of the architecture.
- **Decoupling Regularization**, where we adapt the decoupling regularization loss to this architecture. We introduce the variational estimator $q_\xi(s_t \mid o_t)$ to approximate the task-relevant distribution without conditioning on $c_t$, thereby balancing the conditional dependence within the parallel framework.

Since TransDreamer open-sources its code for the Atari benchmark, we conduct experiments on the first three environments of the Atari 100K benchmark (shown in Figure 17). The results demonstrate that the benefits of leveraging conditional dependence via the collider structure are architecture-agnostic. The proposed methodology is not limited to RSSM-based designs and can effectively enhance transformer-based world models in environments with complex noise interference.

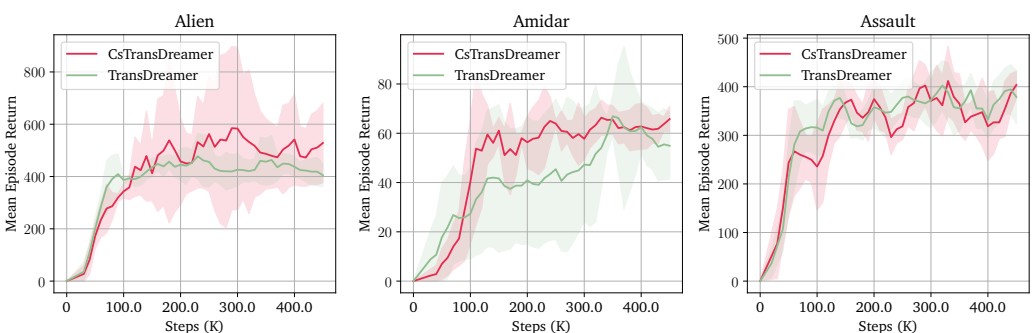

Figure 17: Transformer-based results on three Atari environments.

### E.8 Additional Reconstruction Visualization

We extend the reconstruction visualization to additional environments following the configurations presented in Section 5.4, yielding consistent results (Figure 18-25). Furthermore, in the complex autonomous driving scenario within CARLA (Figure 18), we find that the reconstruction of task-irrelevant variables excluded nearby blue and red vehicles (as shown in the third row). In contrast, the reconstruction of task-relevant variables retains these vehicles, as their proximity influences the decision-making of the autonomous vehicle. These experiments the interpret semantic information extracted by the latent variables and qualitatively demonstrate that our method can effectively extract task-relevant information from noisy observations in complex environments. We also conduct the reconstruction visualization for the train-eval-split experiments in Appendix E.4. In each figure (Figure 26-29), the fourth and fifth rows exhibit a heavily blurred background while maintaining clear task-relevant details. These visualizations demonstrate that CsDreamer effectively learns to extract task-relevant information from noisy observations rather than memorizing the background information, thereby accounting for its superior performance.

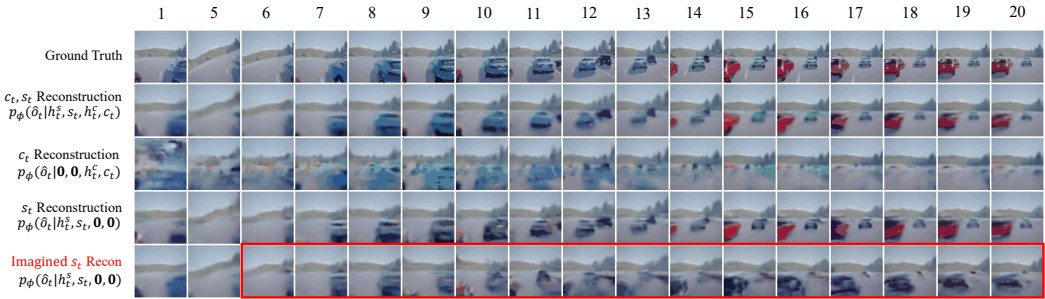

Figure 18: Reconstruction visualization of CARLA

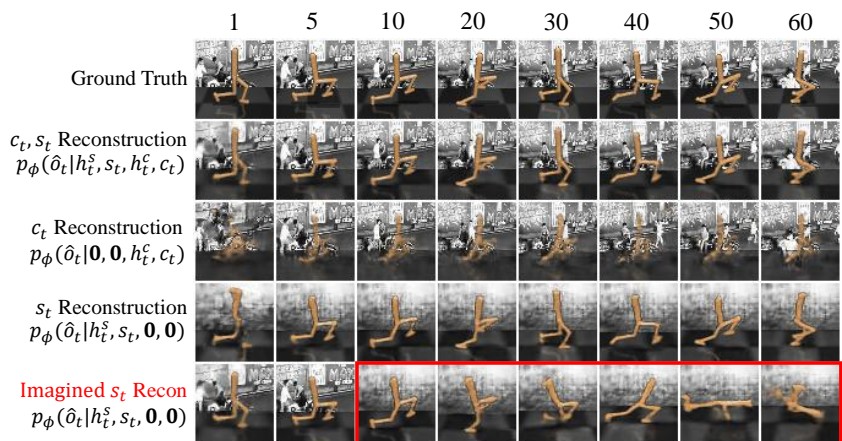

Figure 19: Reconstruction visualization of *Walker Run* using gray videos as background.

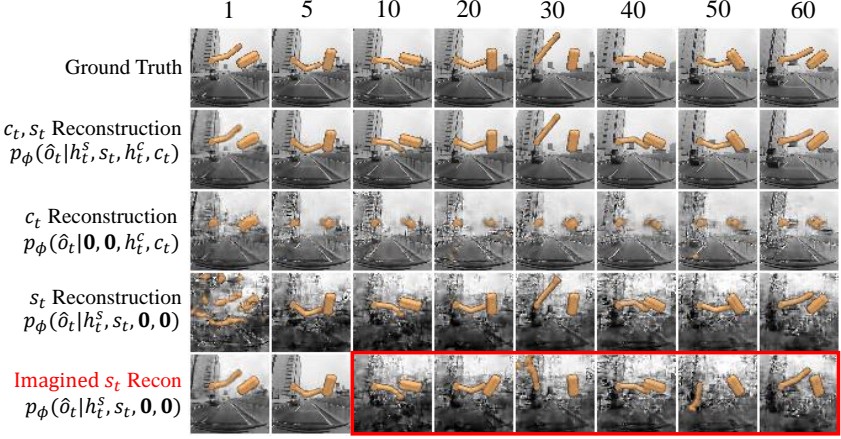

Figure 20: Reconstruction visualization of *Finger Spin* using gray videos as background.

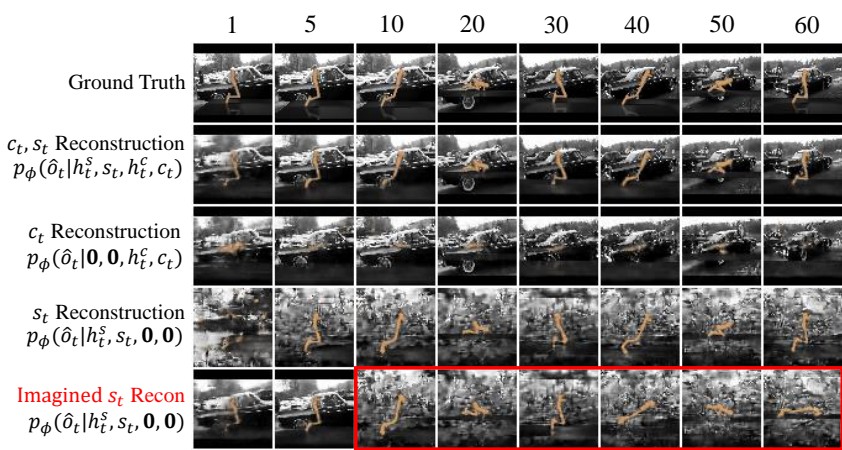

Figure 21: Reconstruction visualization of *Hopper Hop* using gray videos as background.

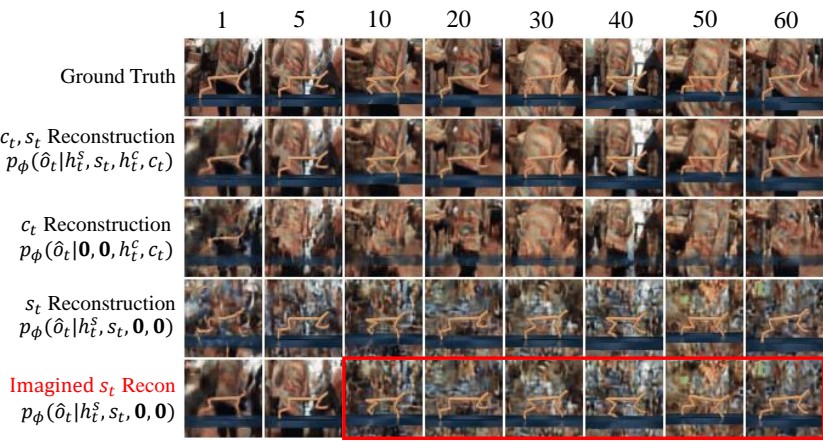

Figure 22: Reconstruction visualization of *Cheetah Run* using colorful videos as background.

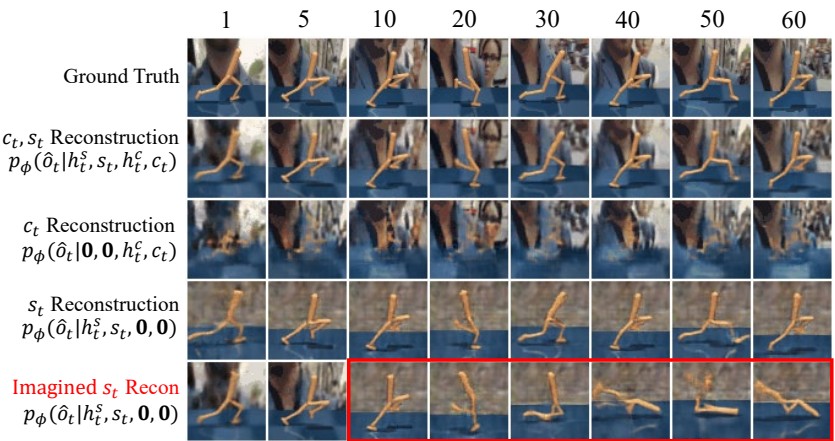

Figure 23: Reconstruction visualization of *Walker Run* using colorful videos as background.

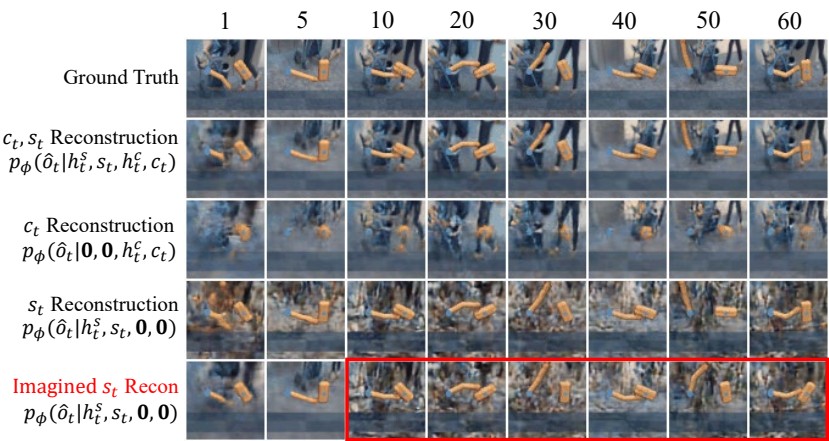

Figure 24: Reconstruction visualization of *Finger Spin* using colorful videos as background.

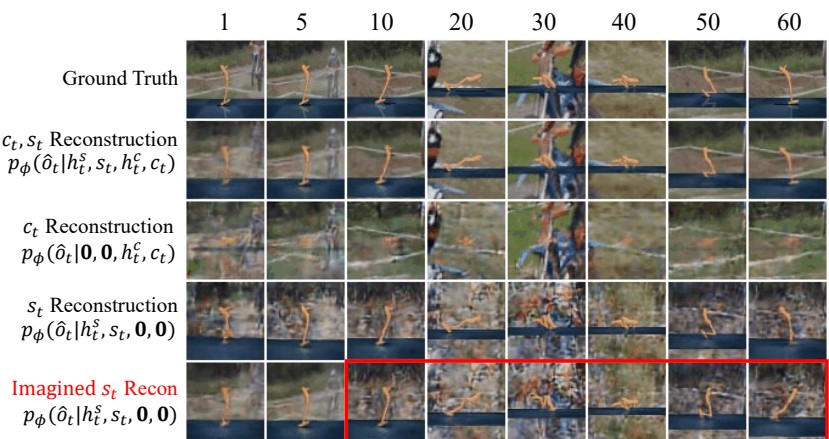

Figure 25: Reconstruction visualization of *Hopper Hop* using colorful videos as background.

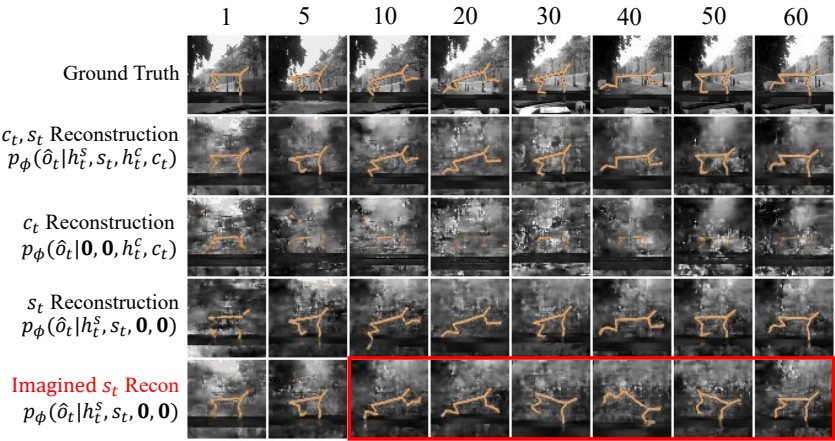

Figure 26: Reconstruction visualization of *Cheetah Run* using train-eval gray videos as background.

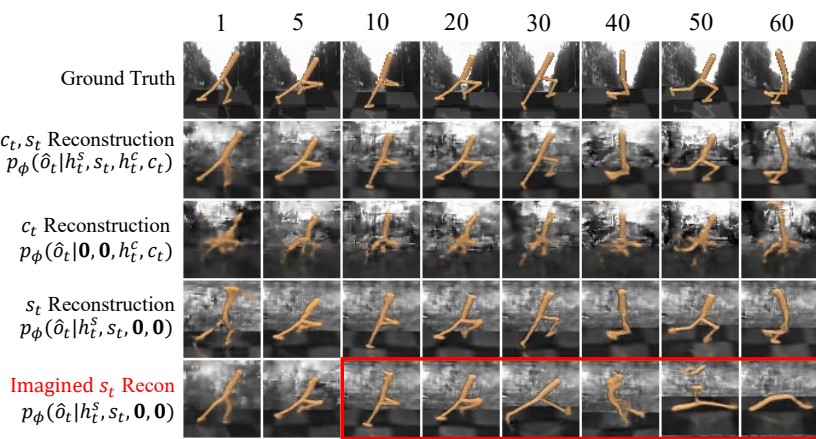

Figure 27: Reconstruction visualization of *Walker Run* using train-eval gray videos as background.

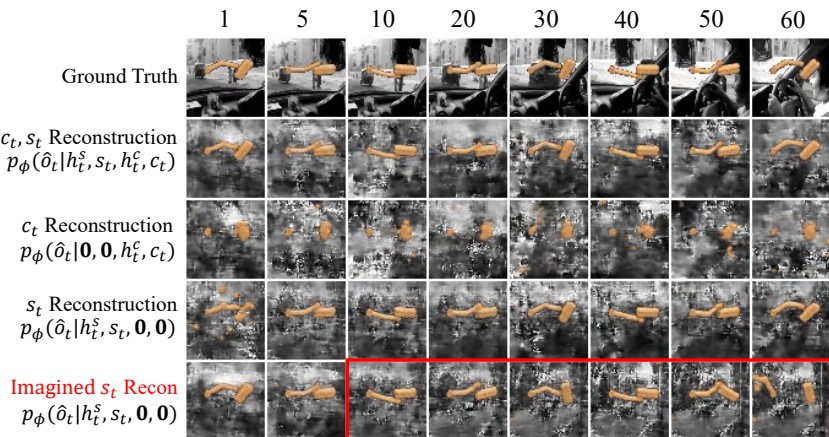

Figure 28: Reconstruction visualization of *Finger Spin* using train-eval gray videos as background.

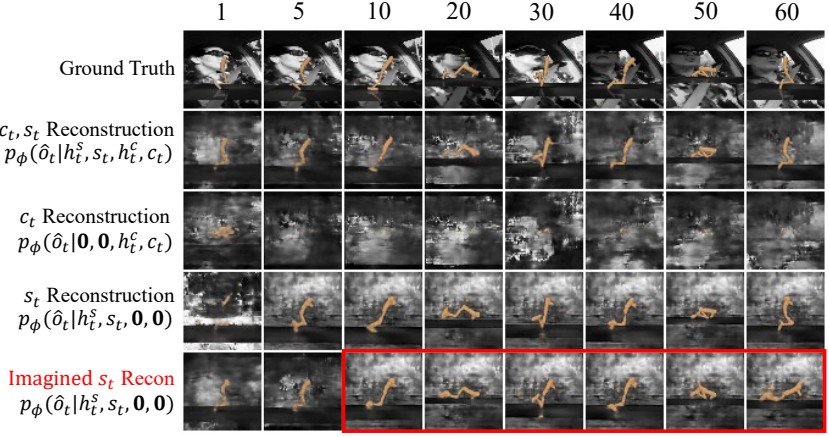

Figure 29: Reconstruction visualization of *Hopper Hop* using train-eval gray videos as background.

