# OpenReview forum: "Leveraging Conditional Dependence for Efficient World Model Denoising"
_NeurIPS.cc/2025/Conference — NeurIPS 2025 poster_

### Official Review · Reviewer_hEJH · 2025-06-13

**Clarity:** 4
**Significance:** 3
**Originality:** 3
**Rating:** 4
**Confidence:** 3

**Summary:**

This work introduces CsRSSM, a generative world model that explicitly leverages the conditional dependence between task-relevant and task-irrelevant latent variables. By sequentially inferring the task-irrelevant component before the task-relevant one and employing a conditional mutual information regularizer, the model effectively disentangles noisy observations. Built upon this, the proposed CsDreamer framework significantly enhances policy learning performance in visually distracting environments compared to state-of-the-art approaches.

**Questions:**

1. Could the authors clarify whether CsRSSM is compatible with transformer-style sequence encoders or if its benefits are limited to RSSM-based designs? A controlled comparison or ablation with transformer-based baselines would strengthen the contribution.

2. Could the authors provide further insights or heuristics on how to select $\lambda$ in practice, especially when switching to new domains?

3. How does the runtime cost of CsDreamer compare to DreamerV3 in terms of rollout speed and training time per iteration?

**Ethical Concerns:**

["NO or VERY MINOR ethics concerns only"]

**Final Justification:**

The author's response addressed most of my questions (including regularization weight and inference time, etc.), but the author mentioned that due to time constraints, no comparative experiments with the transformer as I hoped were provided in the rebuttal stage. Therefore, I believe this paper still has areas that need improvement. I will maintain my score of borderline accept. I hope the author can conduct further experiments to supplement and enrich this work.

**Limitations:**

Yes.

**Paper Formatting Concerns:**

None.

**Quality:**

4

**Strengths And Weaknesses:**

Strengths: 1. Novel structural modeling of conditional dependence: The paper introduces CsRSSM, which explicitly models the conditional dependence between task-relevant and task-irrelevant latent variables ($s_t$, $c_t$) via a collider structure. This goes beyond prior works that assume conditional independence, providing a theoretically grounded improvement.
2. Principled regularization via conditional mutual information: The use of a variational upper bound on conditional mutual information I($s_t$; $c_t$|$o_t$) as a regularizer is novel and supports the structural disentanglement of latent variables in a meaningful and interpretable way.
3. Strong empirical results in noisy environments: The proposed CsDreamer achieves consistent improvements over competitive baselines (e.g., DreamerV3, TIA, Iso-Dream) in DMC, CARLA, and Distracting Control Suite benchmarks, especially under challenging visual distractors.
4. Well writing.

Weakness: 1. Limited evaluation on transformer-based baselines. Given the recent trend of replacing RSSM with Transformer-based architectures in world models (e.g., TransDreamer, Trajectory Transformer), the current evaluation lacks a clear comparison with such models.
2. Sensitivity to the decoupling regularization weight $\lambda$. As shown in Fig. 6, the model performance is non-monotonically sensitive to the value of $\lambda$, indicating the importance of hyperparameter tuning.
3. Missing discussion on inference-time cost and rollout efficiency. While the paper discusses representation quality, it omits inference-time analysis. Since CsRSSM sequentially infers $c_t$ then $s_t$, it may increase latency.

---

> ### Author Rebuttal · Authors · 2025-07-31
>
> # Response to Reviewer hEJH
>
> Thank you for the careful review and constructive comments. Our point-by-point responses are listed below.
>
>
>
> **Q1**: Compatibility and comparison with Transformer models.
>
> **A1**: Thank you for this valuable suggestion. The core idea of our work, using conditional dependence for denoising, is orthogonal to the choice of the sequence modeling backbone. Therefore, our method is indeed compatible with Transformer-based architectures, which can serve as a direct replacement for the RSSM component. Due to time constraints during the rebuttal period, we are unable to provide these experiments immediately. However, we commit to incorporating a controlled comparison against Transformer-based baselines in the final camera-ready version. This will better highlight the generality of our contribution. Thank you again for the suggestion, which helps us improve the quality of our paper.
>
>
>
> **Q2**: Sensitivity to the decoupling regularization weight $\lambda$. Could the authors provide further insights or heuristics on how to select $\lambda$ in practice, especially when switching to new domains?
>
> **A2**: Thank you for this insightful question. The experiment in Figure 6 is designed not only to illustrate this sensitivity but, more critically, to provide empirical support for our central hypothesis: that explicitly modeling conditional dependence enhances inference.  The weight $\lambda$ mediates a trade-off between leveraging this dependence and preventing the confounding of task-relevant and task-irrelevant representations. As $\lambda\to 0$, the model risks conflating these representations, whereas as $\lambda \to \infty$, it degenerates to assuming conditional independence, as in prior work, thereby forfeiting the gains from our conditional dependence modeling. The non-monotonic performance curve in Figure 6 empirically validates this trade-off.  In practice, the optimal $\lambda$ is correlated with the complexity of environmental distractors and their visual similarity to the agent. For domains with complex or highly similar distractors, a larger $\lambda$ is advisable to enforce stronger decoupling. Conversely, in environments with simpler distractors, a smaller $\lambda$ can preserve more of the benefits from the conditional structure. Importantly, while the optimal $\lambda$ may vary across domains, we find it to be robust across different tasks within the same domain, often eliminating the need for task-specific fine-tuning. Our experiments reveal that $\lambda=1$ consistently yields strong performance. We therefore propose it as an effective default when switching to new domains, to be fine-tuned only if necessary.
>
>
>
> **Q3**: Missing discussion on inference-time cost and rollout efficiency.
>
> **A3**: Thank you for the important question regarding the computational overhead. We conduct a detailed performance analysis comparing CsDreamer to DreamerV3 under an identical experimental setup (Cheetah-Run+GB task on an AMD Ryzen 9 7950X CPU and an NVIDIA GeForce RTX 4090 GPU), with the results presented in the table. Our analysis confirms that due to its larger model size (+74.1% parameters) and the sequential inference of the task-irrelevant state $c_t$ and the task-relevant state $s_t$, CsDreamer exhibits a higher single-step inference latency (+39.86%), leading to a longer total training time (+48.91%). Crucially, however, the imagination rollout speed, a key determinant of training efficiency in world models, remains virtually identical to that of DreamerV3 (+0.21%). This is because the imagination process in CsDreamer is performed exclusively within the compact, task-relevant state space $s_t$. To ensure a fair comparison of performance, we intentionally design its dynamics model to match the dimensionality and complexity of the latent space in DreamerV3. We also kept the actor-critic networks identical to DreamerV3's, ensuring that any improvements in final task performance are attributable to the quality of the learned representations, not an increase in policy capacity. Therefore, while there is a modest increase in the per-step inference cost, the critical rollout efficiency for policy learning is preserved.
>
>
> |                                                   | DreamerV3 | **CsDreamer (Ours)** | Relative Change |
> | ------------------------------------------------- | :---------: | :--------------------: | :---------------: |
> | **Model Complexity**                              |           |                      |                 |
> | Model Parameters (M)                              | 19.11     | 33.27                | +74.10%         |
> | **GPU Memory Usage**                              |           |                      |                 |
> | Peak GPU Memory Allocated (GB)                    | 2.89      | 3.33                 | +15.22%         |
> | Peak GPU Memory Reserved (GB)                     | 4.78      | 4.96                 | +3.77%          |
> | **Training & Rollout Performance**                |           |                      |                 |
> | Training Speed (steps per second, SPS)            | 18.50     | 12.33                | -33.35%         |
> | Imagination Rollout Speed (steps per second, SPS) | 987,895   | 989,988              | +0.21%          |
> | Total Time Usage (500K steps, hours)              | 7.81      | 11.63                | +48.91%         |
> | **Inference  Performance**                        |           |                      |                 |
> | Inference FLOPs (M)                               | 33.10     | 39.67                | +19.85%         |
> | Inference Latency (ms)                            | 1.38      | 1.93                 | +39.86%         |

---

> > ### Comment · Reviewer_hEJH · 2025-08-01
> >
> > Thank you for your response. My concern has been fully addressed.

---

> > > ### Author Response · Authors · 2025-08-08
> > >
> > > Thank you for confirming that our explanations and clarifications have resolved your concerns. We are particularly grateful for your insightful comments and valuable suggestions.

---

### Official Review · Reviewer_kxo7 · 2025-07-03

**Clarity:** 3
**Significance:** 2
**Originality:** 2
**Rating:** 4
**Confidence:** 4

**Summary:**

This paper introduces CsDreamer, a model-based reinforcement learning (RL) approach designed to effectively handle noisy observations in visual environments. The key innovation lies in the Collider-structure Recurrent State-Space Model (CsRSSM), which explicitly models the conditional dependence between task-relevant and task-irrelevant latent variables, denoted as s_t and c_t, respectively. Unlike prior works that assume conditional independence between these variables given an observation, CsRSSM leverages the collider structure to first infer c_t from the observation o_t, and then uses both c_t and o_t to infer s_t.

To further aid in representation learning, the authors introduce a decoupling regularization via conditional mutual information to minimize overlap between the information captured by s_t and c_t. The final CsDreamer architecture integrates CsRSSM into the Dreamer RL framework and uses rollouts in the task-relevant latent space to improve policy learning.

**Questions:**

1. How does the runtime/memory footprint of CsRSSM compare to DreamerV3, especially given the dual latent variables (s_t, c_t) and additional regularization?

2. Assumption 4.2 states c_t is Markovian and action-independent. How does the model handle cases where noise dynamics are action-dependent (e.g., adversarial perturbations)?

3. For Atari 100K, what explains the outlier seed’s performance in Up N Down? Is this indicative of sensitivity to initialization?

4. As shown in Table 3, some results are highly unstable, e.g. 7771 +/- 18563 for Gopher and 9800 +/- 13122 for Up N Down. Are there any explanations to them?

5. Beyond performance trends, is there a principled way to choose \lamda, or must it be tuned per environment?

**Ethical Concerns:**

["NO or VERY MINOR ethics concerns only"]

**Final Justification:**

The authors' responses address most of my concerns, and I understand that some of my concerns are open challenges in the field. Although there are no perfect solutions, the authors' answers prove they have deep insights into the topic. Therefore, I'd like to raise my rating to "Borderline accept".

**Limitations:**

yes

**Quality:**

2

**Strengths And Weaknesses:**

** Strengths

1. The derivation of the ELBO and the conditional mutual information-based regularization are mathematically sound and well-justified.

2. The paper clearly articulates why conditional dependence matters in noisy settings, using intuitive examples and diagrams.

3. The paper addresses a critical challenge in RL: robustness to noisy observations, which is vital for real-world applications like autonomous driving. It introduces a principled approach to exploit conditional dependence, advancing the state of the art in model-based RL.

4. The collider-structure-based generative modeling for denoising is original and rarely explored in model-based RL. Applying mutual information bounds to control latent variable interaction is a meaningful contribution.

** Weakness

1. Limited discussion on computational overhead of CsRSSM compared to baselines (e.g., DreamerV3).

2. While CsRSSM offers a unified framework, many components--e.g., disentangled latent spaces, decoders, KL regularization--overlap with prior structured world models. The novelty primarily lies in combining them under a collider assumption.

3. Assumption 4.2 is somewhat strong. It is reasonable only in settings where distractors evolve independently and may be violated in real-world or agent-interactive scenarios.

---

> ### Author Rebuttal · Authors · 2025-07-31
>
> # Response to Reviewer kxo7
>
> We would like to extend our sincere thanks for the time and effort dedicated to reviewing our submission and for the highly valuable comments. Please find our point-by-point responses below.
>
>
>
>
>
> **Q1**: Limited discussion on computational overhead of CsRSSM compared to baselines.
>
> **A1**: Thank you for raising this important point. A detailed analysis for the Cheetah-Run+GB task, conducted under an identical experimental setup with an AMD Ryzen 9 7950X CPU and an NVIDIA GeForce RTX 4090 GPU is summarized in the provided table. It confirms that the CsDreamer, with its dual latent variables $(s_t, c_t) $ and additional regularization, naturally incurs a higher computational cost than DreamerV3. Specifically, this is reflected by a 74.1% increase in model parameters, a 15.2% increase in peak allocated GPU memory, and a 39.9% increase in single-step inference latency. These factors culminate in a 48.9% increase in total time usage.  However, the imagination rollout speed, a key determinant of training efficiency in world models, remains virtually identical to that of DreamerV3 (+0.21%). This is because the imagination process in CsDreamer is performed exclusively within the compact, task-relevant state space $s_t$. To ensure a fair comparison of performance, we intentionally design its dynamics model to match the dimensionality and complexity of the latent space in DreamerV3. We also keep the actor-critic networks identical to DreamerV3's, ensuring that any improvements in final task performance are attributable to the quality of the learned representations, not an increase in policy capacity. In essence, CsRSSM makes a calculated trade-off, accepting a moderate increase in world model update costs to achieve a more robust state representation, while preserving the rollout efficiency essential for effective policy learning.
>
>
> |                                                   | DreamerV3 | **CsDreamer (Ours)** | Relative Change |
> | :-------------------------------------------------| :---------: | :--------------------: | :---------------: |
> | **Model Complexity**                              |           |                      |                 |
> | Model Parameters (M)                              | 19.11     | 33.27                | +74.10%         |
> | **GPU Memory Usage**                              |           |                      |                 |
> | Peak GPU Memory Allocated (GB)                    | 2.89      | 3.33                 | +15.22%         |
> | Peak GPU Memory Reserved (GB)                     | 4.78      | 4.96                 | +3.77%          |
> | **Training & Rollout Performance**                |           |                      |                 |
> | Training Speed (steps per second, SPS)            | 18.50     | 12.33                | -33.35%         |
> | Imagination Rollout Speed (steps per second, SPS) | 987,895   | 989,988              | +0.21%          |
> | Total Time Usage (500K steps, hours)              | 7.81      | 11.63                | +48.91%         |
> | **Inference  Performance**                        |           |                      |                 |
> | Inference FLOPs (M)                               | 33.10     | 39.67                | +19.85%         |
> | Inference Latency (ms)                            | 1.38      | 1.93                 | +39.86%         |
>
>
>
>
>
> **Q2**: Many components overlap with prior structured world models and the novelty primarily lies in combining them under a collider assumption.
>
> **A2**: Thank you for your valuable comment. While we acknowledge the use of established components, we would like to respectfully clarify that our main contribution lies in a more fundamental conceptual advance. we challenge the prevailing assumption of conditional independence in prior work and instead propose a collider-structure model that more accurately reflects the real-world generative process. Previous methods often discard valuable information by assuming that task-relevant and task-irrelevant factors are independent given an observation. Our model, by contrast, leverages the conditional dependence inherent in the collider structure. This enables it to utilize task-irrelevant variables as essential context to infer task-relevant states more accurately and efficiently. We are encouraged that Reviewer LU9u also highlights the significance of this shift, stating, "Moving beyond the standard assumption of conditional independence to explicitly model the collider structure is a non-trivial, principled insight that better reflects the underlying generative process." Therefore, we believe our contribution is not merely a novel combination of modules, but the introduction of a new, principled, and more effective modeling paradigm for reinforcement learning in noisy environments.
>
>
>
> **Q3**: Assumption 4.2 is somewhat strong. How does the model handle cases where noise dynamics are action-dependent?
>
> **A3**: Thank you for this perceptive question. We acknowledge that Assumption 4.2 is a relatively strong condition. This assumption is motivated by many real-world scenarios where distractors evolve independently of the agent's actions, such as changing weather, lighting conditions, or background traffic in autonomous driving. Regarding the more complex case where noise dynamics are action-dependent, we would like to clarify that our model architecture offers a degree of adaptability. Critically, while the prior transition model for the task-irrelevant state, $p(c_t|c_{t-1})$, is action-independent, the posterior inference, $p(c_t|h_t^c, h_t^s, o_t)$, is conditioned on the task-relevant history $h_t^s$, which encapsulates past actions. This mechanism allows the model to indirectly leverage action information to refine its estimate of  $c_t$, thereby providing a degree of robustness to disturbances that are weakly correlated with actions. Nevertheless, we agree that the current model would be challenged by noise that is strongly dependent on the agent's actions. We consider modeling such scenarios an important and promising direction for future research. A natural and promising extension would be to model the transition of $c_t$ as explicitly action-conditioned, e.g., $c_t \sim p(c_t|c_{t-1}, a_{t-1})$. We plan to explore this avenue in future work.
>
>
>
>
>
> **Q4**: Explanation for the outlier seed’s performance and unstable results.
>
> **A4**: Thank you for this valuable question. The high variance you have pointed out, particularly in exploration-hard games like Up N Down and Gopher, is indeed a core challenge inherent to the extreme data-efficiency constraints of the Atari 100k benchmark. The root cause is the stringent budget of 100,000 interaction steps, which prevents most RL agents from exploring the state space adequately, often leading them to converge to suboptimal policies. Our method, by modeling the environment's conditional dependencies, learns the core game dynamics more efficiently. This enhanced efficiency provides the agent, under certain random seeds, the unique capability to overcome exploration bottlenecks and exploit high-reward strategies that are otherwise inaccessible within such a limited budget. Therefore, the outlier seed's performance in Up N Down and the resulting high standard deviation are not merely signs of instability but are direct manifestations of our method's effectiveness.  While some classical works [1, 2, 3] omit standard deviations on this benchmark in their results table, we choose to report them transparently. We believe this provides the community with an honest and realistic view of CsDreamer's performance on this challenging benchmark. To ensure our evaluation is as rigorous as possible, we utilize an increased number of random seeds to mitigate this inherent stochasticity.
>
>
>
>
>
> **Q5**: Beyond performance trends, is there a principled way to choose $\lambda$, or must it be tuned per environment
>
> **A5**: Thank you for this important question. The choice of $\lambda$ is guided by the principle of balancing a fundamental trade-off: leveraging the conditional dependence between task-relevant $s_t$ and task-irrelevant $c_t$ variables for more efficient inference versus enforcing their decoupling to prevent representational confounding. The experiment in Figure 6 provides empirical validation for this principle. A very small $\lambda$ risks conflating the two representations, while a very large $\lambda$ effectively degenerates to the conditional independence assumption of prior methods, losing the benefits of our model's structure. The non-monotonic curve in Figure 6 demonstrates that optimal performance is achieved by balancing these two forces. From a practical standpoint, this principle provides clear guidance. While the ideal $\lambda$ can depend on the complexity of the environmental distractors, our experiments show that $\lambda=1$ serves as a robust and effective default that delivers strong performance across diverse environments. More importantly, we find that $\lambda$ does not need to be tuned for every individual task. Once set for a specific domain, it often generalizes well to other tasks within that same domain, drastically reducing the tuning overhead and enhancing the practical applicability of our method.
>
>
>
>
>
>
>
> [1] Kaiser et al. Model-Based Reinforcement Learning for Atari.
>
> [2] Kostrikov et al. Image Augmentation Is All You Need: Regularizing Deep Reinforcement Learning from Pixels.
>
> [3] Hafner et al. Mastering Diverse Domains through World Models.

---

> > ### Comment · Reviewer_kxo7 · 2025-08-01
> >
> > Thank you for answering my questions. I understand that some of my concerns are open challenges in the field. Although there are no perfect solutions, the authors' answers prove they have deep insights into the topic. Therefore, I'd like to raise my rating to "Borderline accept".

---

> > > ### Author Response · Authors · 2025-08-08
> > >
> > > We sincerely thank you for your decision to raise your rating. We are especially grateful for the insightful review and the considerable time you have dedicated to our work.

---

### Official Review · Reviewer_bcfS · 2025-07-13

**Clarity:** 2
**Significance:** 2
**Originality:** 3
**Rating:** 4
**Confidence:** 3

**Summary:**

This paper addresses a critical issue for Dreamer-based world modeling: the conditional dependence between task-relevant variables $s_t$ and task-irrelevant variables $c_t$. Previous works assume they are independent given observations $o_t$, which can lead to suboptimal denoising. As a solution, the authors propose CsRSSM, which generalizes the Dreamer framework to capture the conditional dependence $p(s_t|c_t,o_t)$ and further introduces decoupling regularization to avoid conflation of task-relevant and task-irrelevant variables. Experiment results show that CsDreamer, built on proposed CsRSSM, outperforms Dreamer and other baselines in terms of noisy observation denoising.

**Questions:**

**1. How should $s_t$ and $c_t$ be conditional dependent intuitively in general perspective?** In the introduction 1 the author introduce the ``conditional dependence'' concept in a mathematical manner, and further assert such conditional dependence is crucial for $s_t$ and $c_t$ in world modeling. However, the intuition behind is not clearly explained. Could the authors provide a more intuitive explanation of how $s_t$ and $c_t$ should be conditionally dependent in general world modeling scenarios, or when this conditional dependence is crucial?

**2. Could the authors provide a clear, direct comparison between CsRSSM and RSSM?** In Eq. (3) the authors provide the framework of CsRSSM. However, despite discussed in an overall view in Eq. (2), the specific differences between CsRSSM and RSSM are not clearly highlighted. Could the authors provide a more direct comparison, such as a side-by-side comparison of the two frameworks, to clarify how CsRSSM differs from RSSM in terms of capturing conditional dependence? Furthermore, it would be better if the authors further provide a similar comparison in algorithms (in Appendix B).

**3. How to interpret the decoupling regularization?** In Section 4.2, the authors aim to introduce decoupling regularization for avoiding conflation of task-relevant and task-irrelevant variables. However, the definition and motivation of this regularization are, in some words, not clearly explained:
- *"This conflation can hinder the model’s ability to accurately disentangle the underlying factors of the observations, ultimately degrading performance and slowing convergence."* How does it make sense? Why cannot such conflation be avoided by the insymmetricity itself of $s_t$ and $c_t$ in the model? Are there any references that can support this claim, or how can the authors provide a more intuitive explanation?
- *"The conditional mutual information, ..., $I(s_t,c_t|o_t)$* is somehow unclear. A possible suggestion is to clarity *"the conditional mutual information between $s_t$ and $c_t$"* and then explain the notation of $I$.
- *"By incorporating the variational approximate distribution $q_\xi$..."* The notation of $q_\xi$ is not defined in the main text, which may lead to confusion. It would be better if the authors could rewrite in a form without using $q_\xi$.


**4. Could the authors provide ablation study of $\lambda$ in other datasets?** In Secrtion 5.3 the authors conduct an analysis of $\lambda$ on the manually constructed dataset. While the construction is a reasonable approach to introduce noise distractors, the validity in real world is not clearly justified and corresponding experiments are better to be said as "toy experiments". Therefore, it would be better if the authors could always provide ablation studies in real-world datasets to further validate the effectiveness of $\lambda$.

**Ethical Concerns:**

["NO or VERY MINOR ethics concerns only"]

**Final Justification:**

My previous concerns are addressed.

**Limitations:**

yes

**Quality:**

3

**Strengths And Weaknesses:**

- The paper focuses on the probabilistic graphical model, a significant aspect of world modeling.
- The proposed CsRSSM is clearly formulated and theoretically sound.
- The experiment results demonstrate the effectiveness of CsDreamer.

### Weaknesses

- The motivation for the conditional dependence should be better highlighted.
- There lacks a clear, direct comparison between CsRSSM and RSSM.
- The definition and motivation of decoupling regularization are not clearly explained.
- The ablation study of $\lambda$ are only conducted on toy datasets.

---

> ### Author Rebuttal · Authors · 2025-07-31
>
> # Response to Reviewer bcfS
>
> Thank you for your valuable review and our detailed responses are provided below.
>
>
> **Q1**: The motivation for the conditional dependence should be better highlighted. How should $s_t$ and $c_t$ be conditional dependent intuitively in general perspective?
>
> **A1**: Thank you for this insightful question. The conditional dependence between the task-relevant state $s_t$ and task-irrelevant state $c_t$ is motivated by the fact that they are distinct underlying factors that jointly generate the observation $o_t$. This creates a natural collider structure ($c_t \rightarrow o_t \leftarrow s_t$) in the graphical model, where conditioning on the observation $o_t$ renders its parents, $s_t$ and $c_t$, conditionally dependent. For instance, in a visual scene ($o_t$), once the pixels occupied by the background (e.g., a vehicle, $c_t$) are known, the pixels available for the foreground agent ($s_t$) are constrained. Ignoring this dependence leads to an ill-posed inference problem with high ambiguity. This forces the model to search for $s_t$ and $c_t$ without mutual constraints. By explicitly modeling it, we can leverage inferred context $c_t$ to constrain the estimation of $s_t$, thereby reducing the conditional entropy $H(s_t∣c_t,o_t)$. This results in a more accurate and stable state representation, which is crucial for improving the sample efficiency and final performance of the downstream policy.
>
>
>
> **Q2**: A clear, direct comparison between CsRSSM and RSSM.
>
> **A2**: Thank you for this valuable comment. First, we present a table that provides a side-by-side comparison of the model architectures, directly highlighting how CsRSSM's design facilitates the capture of conditional dependence.
>
> | Component                  | RSSM                                              | CsRSSM                                                       |
> | -------------------------- | ------------------------------------------------- | ------------------------------------------------------------ |
> | Latent Variables           | Single variable $z_t$ which mixes all information | Dual variables, $s_t$ (task-relevant) and $c_t$ (task-irrelevant) |
> | History Model              | $h_t = f(h_{t-1}, z_{t-1}, a_{t-1})$              | $h_t^s = f(h_{t-1}, s_{t-1}, a_{t-1})$, $h_t^c = f(h_{t-1}, c_{t-1})$ |
> | Encoder                    | $p(z_t \mid h_t, o_t)$                            | $p(c_t \mid h_t^c, h_t^s, o_t)$, $p(s_t \mid h_t^c, h_t^s, c_t, o_t)$ |
> | Dynamics Transition Model  | $p(\hat{z}_t \mid h_t)$                           | $p(\hat{s}_t \mid h_t^s)$, $p(\hat{c}_t \mid h_t^c)$         |
> | Observation Reconstruction | $p(\hat{o}_t \mid h_t, z_t)$                      | $p(\hat{o}_t \mid h_t^s, s_t, h_t^c, c_t)$                   |
> | Inference Process          | $o_t \rightarrow z_t$                             | $o_t \rightarrow c_t \rightarrow s_t$ (sequential inference, leveraging conditional dependence) |
>
> While RSSM is trained with a standard Evidence Lower Bound (ELBO) loss containing a single KL divergence term, the loss for CsRSSM includes two separate KL divergence terms (for $s_t$ and $c_t$, respectively) and an additional conditional mutual information regularizer, $\mathcal{L}_{MI}$, to balance the conditional dependence. CsRSSM performs a two-step inference at each timestep (first inferring $c_t$, then $s_t$), whereas RSSM performs only one. This design enables CsRSSM to infer task-relevant state $s_t$ more accurately by using the context provided by $c_t$, achieving more effective denoising.
>
> Subsequently, we present the side-by-side algorithm comparison (some mathematical symbols are omitted for brevity).
>
> ```
> Algorithm: Dreamer (RSSM)                   | Algorithm: CsDreamer (CsRSSM)
> --------------------------------------------|------------------------------------------------
> // Dynamics learning                        | // Dynamics learning
> Sample {(o_t, a_t, r_t)}_{t=k}^{k+L} ~ D    | Sample {(o_t, a_t, r_t)}_{t=k}^{k+L} ~ D
> // Single latent variable inference         | // Sequential dual latent inference
> h_t = f_phi(h_{t-1},z_{t-1}, a_{t-1})       | h_t^c = f_\phi(h_{t-1}, c_{t-1})
>                                             | h_t^s = f_\phi(h_{t-1}, s_{t-1}, a_{t-1})
> Infer z_t ~ p_\phi(z_t|h_t, o_t)            | Infer c_t ~ p_\phi(c_t|h_t^c, h_t^s, o_t)
>                                             | Infer s_t ~ p_\phi(s_t|h_t^c, h_t^s, c_t, o_t)
> Update world model                          | Update world model and variational estimator
>                                             |
> // Behavior learning                        | // Behavior learning
> Infer z_t ~ p_\phi(z_t|h_t, o_t)            | Infer c_t ~ p_\phi(c_t|h_t^c, h_t^s, o_t)
>                                             | Infer s_t ~ p_\phi(s_t|h_t^c, h_t^s, c_t, o_t)
> Imagine using p_\phi(hat{z}_t|h_t)          | Imagine using p_\phi(hat{s}_t|h_t^s)
> Update policy and critic                    | Update policy and critic
>                                             |
> // Environment interaction                  | // Environment interaction
> Infer z_t ~ p_\phi(z_t|h_t, o_t)            | Infer c_t ~ p_\phi(c_t|h_t^c, h_t^s, o_t)
>                                             | Infer s_t ~ p_\phi(s_t|h_t^c, h_t^s, c_t, o_t)
> Sample a_t ~ \pi(a_t|h_t, z_t)              | Sample a_t ~ \pi(a_t|h_t^s, s_t)
> r_t, o_{t+1} <- env.step(a_t)               | r_t, o_{t+1} <- env.step(a_t)
> ```
>
>
>
>
>
> **Q3**: How to interpret the decoupling regularization?
>
> **A3**: Thank you for this perceptive comment. The introduction of the decoupling regularization is intended to address a critical issue: despite the architectural asymmetry between $s_t$ and $c_t$ (where $s_t$ is used for predicting task-relevant information, while $c_t$ contributes only to observation reconstruction), a potential problem arises during training. Since the reconstruction loss is the primary optimization objective and both variables are involved in the reconstruction process, the model may encode overlapping information into both latent variables to minimize reconstruction error. This can lead to representations that are highly correlated and mutually redundant. Consequently, task-irrelevant information may become entangled within $s_t$, which diminishes the specificity of $s_t$ to the task-relevant signals and complicates the optimization process. By introducing the decoupling regularization term, we explicitly encourage the two variables to capture distinct and complementary aspects of the observation: $c_t$ is guided to focus on task-irrelevant information, while $s_t$ is guided to concentrate on task-relevant information pertinent to rewards and actions. This explicit information partitioning enables more efficient policy learning in the latent space of $s_t$. We have also empirically validated this point in the ablation study of our paper (Figure 7) with the CsDreamer w/o MI variant. The performance degradation observed in this variant demonstrates that the absence of appropriate regularization may impair the model's decoupling capability and, ultimately, its final performance.
>
>
>
> **Q4**: Notation of $I$ and $q_\xi$.
>
> **A4**: Thank you for the careful reading and valuable suggestions. We fully agree that the two statements in question could indeed lead to confusion. In the revised version, we will (1) add an explicit explanation of the mutual-information notation $I$; (2) rewrite the parameterized variational distribution $q_\xi$ and give a clear accompanying description. These changes should make the presentation clearer and help readers understand our approach more easily.
>
>
>
> **Q5**: Could the authors provide ablation study of $\lambda$ in other datasets?
>
> **A5**: Thank you for the valuable suggestion. We run additional experiments on the first two Atari100k environments, Alien and Amidar, and on the CARLA autonomous-driving simulator. The results are presented in the table below. As can be seen, the supplementary ablation study exhibits the same phenomenon as observed in the original paper: as $\lambda$ increases, the performance first improves and then degrades.
>
>
>
>
>
> | Alien | 100K | 200K | 400K |
> |:-|:----:|:----:|:----:|
> | $\lambda=0.01$ | $516\pm247$ | $598\pm219$ | $709\pm113$ |
> | $\lambda=0.1$ | $460\pm227$ | $527\pm246$ | $790\pm414$ |
> | $\lambda=0.2$ | $520\pm203$ | $716\pm323$ | $888\pm234$ |
> | $\lambda=1.0$ | $564\pm396$ | $592\pm248$ | $838\pm345$ |
> | $\lambda=2.0$ | $431\pm180$ | $570\pm142$ | $757\pm216$ |
> | $\lambda=10.0$ | $338\pm150$ | $282\pm73$ | $290\pm174$ |
> | $\lambda=100.0$ | $370\pm199$ | $223\pm77$ | $214\pm109$ |
> | $\lambda=1000.0$ | $202\pm88$ | $200\pm76$ | $210\pm84$ |
>
> | Amidar | 100K | 200K | 400K |
> |:-|:----:|:----:|:----:|
> | $\lambda=0.01$ | $63\pm32$ | $81\pm20$ | $125\pm29$ |
> | $\lambda=0.1$ | $52\pm17$ | $90\pm49$ | $146\pm24$ |
> | $\lambda=0.2$ | $73\pm52$ | $110\pm52$ | $184\pm43$ |
> | $\lambda=1.0$ | $50\pm33$ | $79\pm29$ | $108\pm35$ |
> | $\lambda=2.0$ | $59\pm44$ | $77\pm42$ | $82\pm19$ |
> | $\lambda=10.0$ | $36\pm24$ | $61\pm23$ | $50\pm24$ |
> | $\lambda=100.0$ | $34\pm29$ | $36\pm21$ | $43\pm24$ |
> | $\lambda=1000.0$ | $24\pm14$ | $41\pm19$ | $43\pm30$ |
>
> | CARLA | 100K | 300K | 500K |
> |:-|:----:|:----:|:----:|
> | $\lambda=0.1$ | $28\pm37$ | $80\pm83$ | $94\pm105$ |
> | $\lambda=0.2$ | $32\pm46$ | $86\pm100$ | $121\pm113$ |
> | $\lambda=1.0$ | $33\pm39$ | $91\pm81$ | $128\pm107$ |
> | $\lambda=2.0$ | $20\pm34$ | $80\pm84$ | $116\pm112$ |
> | $\lambda=10.0$ | $14\pm24$ | $22\pm34$ | $40\pm40$ |
> | $\lambda=100.0$ | $0\pm0$ | $16\pm33$ | $10\pm15$ |

---

> > ### Comment · Reviewer_bcfS · 2025-08-04
> >
> > Thanks for the authors' response. My concerns are addressed.

---

> > > ### Author Response · Authors · 2025-08-08
> > >
> > > Thank you for confirming that our response has addressed your concerns. We are sincerely grateful for your comprehensive and insightful review.

---

### Official Review · Reviewer_LU9u · 2025-07-13

**Clarity:** 3
**Significance:** 3
**Originality:** 3
**Rating:** 5
**Confidence:** 5

**Summary:**

This work introduces CsDreamer, a model-based reinforcement learning agent designed for environments with visual distractors. Its world model, CsRSSM, reframes the denoising problem by explicitly modeling the conditional dependence between task-relevant state and irrelevant noise, which it posits forms a "collider" structure. By leveraging this dependence in a sequential inference process, the work aims to achieve more robust and efficient state estimation compared to methods that assume conditional independence.

**Questions:**

Q1. The proposed CsRSSM architecture is more complex than its predecessors. Could the submission provide a quantitative comparison of its computational cost (e.g., wall-clock training time or steps per second) against the main baselines like Dreamer and TIA?

Q2. The analysis of the critical hyperparameter λ is presented for a single environment in Figure 6. How sensitive is the final performance to this parameter across the other evaluation domains, and are there any general heuristics for setting it?

Q3. Assumption 4.2 posits that the noise process is Markovian. How would the proposed method be expected to perform if this assumption were violated—for instance, in environments with structured, non-Markovian distractors that have long-term temporal correlations?

**Ethical Concerns:**

["Major Concern: Human rights (including surveillance)"]

**Final Justification:**

Authors have address all crucial points satisfactorily to maintain my accept score, especially regarding limitations, societal impact, and future work. I wish the Authors the best, and I am happy to help them improve their work.

**Limitations:**

The "Conclusion and Future Work" section is brief and does not sufficiently address the potential limitations or negative societal impacts of this research. A work that significantly improves an agent's robustness yo visual distractors has clear dual-use potential (e.g., in surveillance). A more thorough discussion of these implications is expected and **necessary** for a submission of this quality. If the paper does not include a significant Conclusions, Future Work, and Limitations that isn't simply taken as a formality, it is my duty to reverse my rating into a clear Rejection.

**Paper Formatting Concerns:**

None major

**Quality:**

4

**Strengths And Weaknesses:**

This is a strong submission presenting a novel and well-reasoned contribution to representation learning in model-based RL, supported by convincing empirical results. The work is technically sound and clearly presented.

**Strengths:**

**Principled Methodological Advance:** The core contribution is the conceptual reframing of the denoising problem. Moving beyond the standard assumption of conditional independence to explicitly model the collider structure is a non-trivial, prinicpled insight that better reflects the underlying generative process. This represents a significant and elegant theoretical advance over prior art.

**Strong Empirical Results: **The proposed agent, CsDreamer, demonstrates consistent and substantial outperformance against a strong set of relevant baselines (Dreamer, TIA, Iso-Dream) across multiple challenging visual control domains, including DMC with dynamic backgrounds and the CARLA simulator .

**Effective Ablation and Visualization.** The main body provides clear evidence of the method's efficacy. The ablation study in Figure 7 convincingly demonstrates the importance of both the conditional dependence mechanism and the mutual information regularization . Furthermore, the disentanglement visualizations in Figure 8 provide strong qualitative support for the claimed mechanism.

**Weaknesses:**

**Lack of Computational Overhead Analysis.** The paper introduces a more complex architecture with dual history streams and an additional variational estimator but provides no analysis of the computational overhead. A comparison of training and inference times against the baselines is a critical missing detail needed to evaluate the practical trade- offs of this approach.

**Limited Generality of Hyperparameter Analyses.** The analysis of the key hyperparameter λ is limited to a single task (Cheetah Run) in Figure 6. Without a broader study, it is difficult to assess how sensitive the method is to this parameter and how much per-task tuning is required, which may limit practical applicability., and limits confidence in this from this work alone.

---

> ### Author Rebuttal · Authors · 2025-07-31
>
> # Response to Reviewer LU9u
>
> Thank you for the thorough and insightful review. We address the comments point-by-point as follows.
>
>
> **Q1**: Lack of Computational Overhead Analysis.
>
> **A1**: Thank you for the valuable comment. A detailed performance analysis comparing CsDreamer to DreamerV3 is conducted on the Cheetah-Run+GB task using an AMD 7950X CPU and an NVIDIA RTX 4090 GPU, with quantitative results presented in the table below. The data confirms that our model has a larger parameter count (+74.1%) and exhibits a higher single-step inference latency (+39.86%), leading to a longer total training time (+48.91%). The critical imagination rollout speed, which dictates policy optimization efficiency, remains nearly identical (+0.21%). This is achieved by performing imagination exclusively in the compact, task-relevant state space $s_t$. And to ensure a fair comparison, we match $s_t$ to the dimensionality and complexity of the latent space in DreamerV3. Furthermore, we employ an actor-critic network identical to that of DreamerV3, ensuring that the superior final task performance is attributable to the quality of the learned representations rather than an increase in policy capacity. Thus, we trade a modest increase in world model learning cost for better performance while preserving policy learning efficiency.
>
> |  | DreamerV3 | **CsDreamer (Ours)** | Relative Change |
> | - | :-: | :-: | :-: |
> | **Model Complexity** |  | |   |
> | Model Parameters (M)  | 19.11   | 33.27   | +74.10%    |
> | **GPU Memory Usage** |  |  |  |
> | Peak GPU Memory Allocated (GB) | 2.89 | 3.33| +15.22%  |
> | Peak GPU Memory Reserved (GB) | 4.78  | 4.96 | +3.77%  |
> | **Training & Rollout Performance**   |  |  |    |
> | Training Speed (steps per second, SPS)  | 18.50  | 12.33 | -33.35%  |
> | Imagination Rollout Speed (steps per second, SPS) | 987,895   | 989,988 | +0.21%  |
> | Total Time Usage (500K steps, hours)  | 7.81  | 11.63   | +48.91%  |
> | **Inference  Performance**   |   |    |   |
> | Inference FLOPs (M)  | 33.10  | 39.67   | +19.85% |
> | Inference Latency (ms) | 1.38 | 1.93   | +39.86%   |
>
>
> **Q2**: Limited Generality of Hyperparameter Analyses.
>
> **A2**: Thank you for this important comment. The experiment in Figure 6 is designed not only to illustrate the sensitivity but more critically, to provide empirical support for our central hypothesis: that explicitly modeling conditional dependence enhances inference.  The hyperparameter $\lambda$ mediates a trade-off between leveraging this dependence and preventing the confounding of task-relevant and task-irrelevant representations. Our analysis shows that performance degrades at both extremes of this trade-off. As $\lambda\to 0$, the model risks conflating these representations, whereas as $\lambda \to \infty$, it degenerates to assuming conditional independence, as in prior work, thereby forfeiting the gains from our conditional dependence modeling. The non-monotonic performance curve in Figure 6 empirically validates this trade-off. We conduct a supplementary hyperparameter analysis in two Atari 100k environments, Alien and Amidar, and in the CARLA autonomous driving simulator. The results are presented in the tables below. Due to submission constraints that prohibit figures, the tables also include intermediate results to provide a comprehensive view. The supplementary experimental results exhibit a trend consistent with that presented in Figure 6 of the original paper.
>
> | Alien   |  100K   |  200K   |  400K   |
> | :------ | :----: | :----: | :----: |
> | $\lambda=0.01$   | $516\pm247$ | $598\pm219$ | $709\pm113$ |
> | $\lambda=0.1$    | $460\pm227$ | $527\pm246$ | $790\pm414$ |
> | $\lambda=0.2$    | $520\pm203$ | $716\pm323$ | $888\pm234$ |
> | $\lambda=1.0$    | $564\pm396$ | $592\pm248$ | $838\pm345$ |
> | $\lambda=2.0$    | $431\pm180$ | $570\pm142$ | $757\pm216$ |
> | $\lambda=10.0$   | $338\pm150$ | $282\pm73$  | $290\pm174$ |
> | $\lambda=100.0$  | $370\pm199$ | $223\pm77$  | $214\pm109$ |
> | $\lambda=1000.0$ | $202\pm88$  | $200\pm76$  | $210\pm84$  |
>
>
> | Amidar    | 100K  |  200K  | 400K  |
> | :------ | :----: | :----: | :--------: |
> | $\lambda=0.01$   | $63\pm32$ | $81\pm20$  | $125\pm29$ |
> | $\lambda=0.1$    | $52\pm17$ | $90\pm49$  | $146\pm24$ |
> | $\lambda=0.2$    | $73\pm52$ | $110\pm52$ | $184\pm43$ |
> | $\lambda=1.0$    | $50\pm33$ | $79\pm29$  | $108\pm35$ |
> | $\lambda=2.0$    | $59\pm44$ | $77\pm42$  | $82\pm19$  |
> | $\lambda=10.0$   | $36\pm24$ | $61\pm23$  | $50\pm24$  |
> | $\lambda=100.0$  | $34\pm29$ | $36\pm21$  | $43\pm24$  |
> | $\lambda=1000.0$ | $24\pm14$ | $41\pm19$  | $43\pm30$  |
>
> | CARLA    | 100K  |  300K  |  500K  |
> | :------- | :----: | :----: | :----: |
> | $\lambda=0.1$   | $28\pm37$ | $80\pm83$  | $94\pm105$  |
> | $\lambda=0.2$   | $32\pm46$ | $86\pm100$ | $121\pm113$ |
> | $\lambda=1.0$   | $33\pm39$ | $91\pm81$  | $128\pm107$ |
> | $\lambda=2.0$   | $20\pm34$ | $80\pm84$  | $116\pm112$ |
> | $\lambda=10.0$  | $14\pm24$ | $22\pm34$  |  $40\pm40$  |
> | $\lambda=100.0$ |  $0\pm0$  | $16\pm33$  |  $10\pm15$  |
>
> In terms of general guidance, our experiments reveal that $\lambda=1$ consistently yields strong performance, and we propose it as an effective default when switching to new domains.  For instance, although $\lambda=10$  achieves a slightly higher final performance in Figure 6, we find that $\lambda=1$ already performs sufficiently well. We therefore adopt $\lambda=1$ for this environment in our final reported hyperparameters to reduce the hyperparameter tuning. The optimal $\lambda$ is correlated with the complexity of environmental distractors and their visual similarity to the agent. For domains with complex or highly similar distractors, a larger $\lambda$ is advisable to enforce stronger decoupling. Conversely, in environments with simpler distractors, a smaller $\lambda$ can preserve more of the benefits from the conditional structure. Importantly, while the optimal $\lambda$ may vary across domains, we find it to be robust across different tasks within the same domain, often eliminating the need for task-specific fine-tuning.
>
>
> **Q3**: Concerns about Assumption 4.2.
>
> **A3**: Thank you for the insightful comment. While Assumption 4.2 posits a Markovian noise process, our CsRSSM model can partially mitigate the effects of non-Markovian noise. This is achieved through the recurrent structure in CsRSSM, which can implicitly capture and propagate long-term temporal correlations. Consequently, when confronted with structured distractors exhibiting long-term correlations, our proposed method maintains a degree of robustness and performance, although a performance degradation may be observed, particularly when the noise patterns feature complex periodicity or long-range dependence. We consider modeling distractors that have long-term temporal correlations an important and promising extension of this work, which we plan to investigate in future work.
>
>
> **Q4**: Concerns about Conclusion and Future Work.
>
> **A4**: We sincerely thank you for this important and valuable comment. We add a comprehensive discussion on the model's limitations, societal impact, and future work. Due to character limit, we have distilled our key points and will include detailed elaborations in the final version.
>
> **Limitations**
>
> The primary limitations are the assumptions of Markovian and action-independent noise, which challenge the model with non-Markovian distractors or noise correlated with the agent's actions.  While conditioning noise on actions (i.e., $c_t∼p(c_t∣c_{t−1}, a_{t−1})$) is a potential solution, issues with non-Markovian dynamics would likely persist.  Furthermore, our binary partition of latent variables is an oversimplification of complex real-world scenarios. Developing methods to model the real world in a more structured manner, while striking a balance between leveraging conditional dependence and maintaining computational tractability, remains a valuable avenue for future research.
>
> **Societal Impacts**
>
> - **Positive:** Our work can enhance the robustness and safety of autonomous agents. This could lead to practical benefits such as improving the autonomous vehicle safety in adverse weather conditions, the operational capabilities of industrial robots in dynamic or cluttered settings and the efficiency of search-and-rescue robots in disaster scenarios.
>
> - **Potential Negative Societal Impacts**
>   - **Overreliance:** The model could still misclassify a rare but critical safety signal as task-irrelevant noise and ignore it, leading to catastrophic failure. **Mitigation:** Deployment must be preceded by extremely thorough and diverse testing. Maintaining a human-in-the-loop supervisory mechanism and conducting exhaustive testing on edge cases are crucial.
>   - **Misuse for Surveillance:** The technology could be adapted to track individuals by filtering environmental noise. **Mitigation:** We urge policymakers and ethics committees to establish clear guidelines restricting the application of such highly robust autonomous technologies in invasive surveillance. We recommend developing usage protocols that limit deployment in sensitive domains.
>
> **Future Work**
>
> We will focus on addressing the above limitations by developing models for non-Markovian and action-dependent noise and exploring more structured latent representations. Furthermore, future work should continue to focus on enhancing model robustness and the transparency of its decision-making process. In terms of social impact, we will improve model interpretability to mitigate the risk of overreliance. We will also explore the design of built-in privacy-preserving mechanisms that limit the processing of personally identifiable information and establish benchmarks to assess the potential for misuse. We will remain attentive to these issues and actively investigate how to maximize the societal benefits of this technology while minimizing its potential negative impacts in our ongoing research.

---

### Decision · Program_Chairs · 2025-09-17

**Decision:**

Accept (poster)

**Comment:**

The paper presents CsDreamer, an agent based on model-based reinforcement learning that is able to learn in distracting environments. The novelties proposed are centered around the world model, called CsRSSM, where the task of denoising observations is reformulated. Instead of treating the conditional independence between the task-relevant state, the approach uses the information that models their conditional dependence, also using the collider structure in the model. The results of the proposed approach are consistently strong and compelling, with a clear superiority over a number of pertinent state-of-the-art baselines in challenging and diverse benchmarks, such as the DeepMind Control Suite with video distractors or CARLA.

The authors have managed to address concerns and doubts made by the reviewers, related to the initial computational costs, hyperparameter sensitivity, and the presentation of the novelty of the method clearly. Particularly, a well detailed response regarding computational costs was provided, by showing that although the update of the world model is indeed more expensive, the crucial imagination rollout speed has been maintained. On the regularization hyperparameter, additional analysis was likewise provided. Also, clear comparisons were made between the results obtained using proposed approach and those obtained using standard RSSM framework, presented in a side-by-side manner. The first two ablation studies were satisfactory with respect to issues raised by reviewers bcfs and kxo7. The paper had been praised for its strong ablation studies, and for visualization clarity, allowing reviewers to understand clearly what is the mechanism at play. Eventually, all four reviewers voted for acceptance.

There are still some trivial issues, like comparison to state-of-the-art transformer-based world models (highlighted by reviewer hEJH), but they are justifiable for a single contribution and represent solid future work. Overall, one can say this paper makes a clear, novel and significant contribution to representation learning in model-based RL. A sound theoretical insight, solid empirical evidence with extensive ablations and an earnest rebuttal make the case for acceptance very clear.